# METAPERCH: Learning from Metadata for Bioacoustics Foundation Models

**Mustafa Chasmai** [1] [†] **Vincent Dumoulin** [2] **Jenny Hamer** [2]

## Abstract

Bioacoustic foundation models rely on large-scale citizen science platforms like Xeno-Canto for geographically and ecologically diverse data. Recent work has shown that supervision alone can produce SotA species detection models when trained on this large-scale data—however, there remains unutilized potential in the form of recording metadata readily available within these community-driven data hubs. In this work, we explore the use of metadata—such as location and time—as auxiliary supervision signals, allowing the model to leverage species-metadata correlations in its learned representation. Auxiliary metadata losses provide additional information beyond vocalizations alone that can encourage a richer, more robust representation that generalizes better to species distribution and acoustic domain shifts—important challenges for deployment in real-world passive acoustic monitoring (PAM) settings. We introduce META-PERCH, a new foundation model that achieves strong species identification performance across multiple challenging domains and present an extensive empirical study of the effects of 9 diverse metadata sources on 17 bioacoustic datasets.

## 1. Introduction

Bioacoustics is an interdisciplinary field that aims to understand the natural world through sound. Advanced machine learning (ML) techniques have been increasingly used in bioacoustics (Stowell, 2022), particularly for species identification. ML and AI models help scale biodiversity monitoring by automating species identification and saving significant manual effort. Fine-grained differences in species sounds, limited data, and long-tailed distributions present unique challenges for AI research. Recent progress

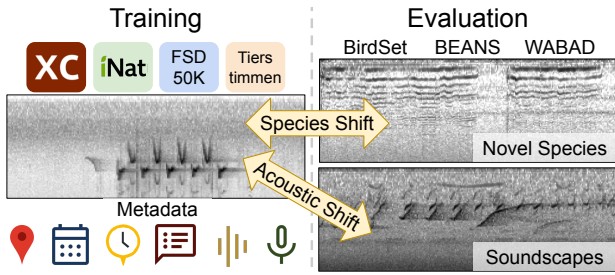

*Figure 1.* **Problem setup**. Bioacoustics training and evaluation settings differ significantly. This figure illustrates substantial acoustic (focal → soundscape) and species (observed → novel) domain shifts in our setup. In this work, we explore the use of metadata as auxiliary supervision to learn robust, generalizable features and help bridge the domain gap.

has been catalyzed by new datasets and evaluation benchmarks (Rauch et al., 2025b; Hagiwara et al., 2023; Chasmai et al., 2024), giving rise to bioacoustic foundation models (Schwinger et al., 2025; van Merriënboer et al., 2025).

Model development in this field is supported by large-scale global platforms such as Xeno-Canto (Xeno Canto) and iNaturalist (iNaturalist). By allowing citizen scientists to upload and annotate recordings from everyday devices like mobile phones, these platforms have facilitated the collection of sounds of diverse taxa from around the world. Citizen science recordings tend to be curated, focal recordings, often clustered around human population centers, and often featuring popular species. In contrast, biodiversity monitoring applications tend to use soundscapes that are collected from continuous, passive acoustic monitoring (PAM) devices and have much more background noise and species overlap. Such efforts also typically target ecologically important or endangered zones such as tropical rainforests or wetlands and give considerable importance to rare species. This disparity between the training data and deployment targets means that bioacoustics models must be robust to challenging domain shifts: acoustic, species presence/distribution, and geographic (Figure 1).

Despite the drawbacks, the sheer volume of annotated data provided by these platforms is an essential component of recent bioacoustic foundation models. Furthermore, citizen science websites include a wealth of auxiliary information such as recording location, date, and quality; individ-

---

[1]University of Massachusetts Amherst. [†]Work done as a student researcher at Google DeepMind. [2]Google DeepMind. Correspondence to: Jenny Hamer <hamer@google.com>.

*Proceedings of the 43rd International Conference on Machine Learning*, Seoul, South Korea. PMLR 306, 2026. Copyright 2026 by the author(s).

ual call type, sex, and life stage; free-form notes left by the recordist as well as certain background species and recording device characteristics. Global taxonomic patterns in these metadata sources could be utilized to learn useful correlations that help bridge the domain gaps. While recent work has demonstrated the utility of supervised learning (van Merriënboer et al., 2025), it often uses species annotations as the only source of supervision. In this work, we employ a multi-task learning approach, jointly training our model with the different metadata prediction tasks alongside the primary species identification task, and quantify the effects of learning from metadata in this setting.

Multi-task learning with metadata presents several key challenges. First, complete coverage of all metadata is often not practical and models must appropriately handle missing metadata. Second, popular data augmentations such as *mixup* (Zhang et al., 2018) require care when mixing recordings with varying metadata availabilities. Third, as we include more metadata, the number of training losses increases and the multi-objective optimization problem becomes more complex. Lastly, some metadata sources may encourage spurious correlations not grounded in ecological interpretation, potentially worsening the domain gap. In this work, we tackle these challenges to build metadata-aware species identification models.

We summarize our main contributions below:

1. We propose METAPERCH[1], a new bioacoustic foundation model that uses metadata as auxiliary targets.
2. We demonstrate that learning with metadata improves species identification with extensive experiments across a wide range of datasets and tasks covering diverse taxa and geographical regions.
3. We present a comprehensive empirical study of the disentangled importance of different metadata modalities and the effects of various loss formulations and other design choices in our training paradigm.

## 2. Related work

**Conditioning on metadata.** At test-time, metadata can provide additional context for species identification. Apps such as Merlin Sound ID use location to help disambiguate between acoustically similar species. Chasmai et al. (2024) use species range estimation models (Cole et al., 2023) to construct geographic priors (Mac Aodha et al., 2019) and constrain the set of possible species at each location. Kahl et al. (2021) use location and week-of-year to train their own species range model for similar test-time species filtering. Jeantet & Dufourq (2023) also explore the benefits of location and time as inputs or priors. Robinson et al. (2024a) explore the inclusion of location and recordist

notes as free form text for zero-shot captioning. A key assumption here is the availability of metadata at test-time. In contrast, our work uses metadata solely during training, making it easier to handle missing metadata and facilitating broader downstream applicability.

**Training with metadata.** The use of metadata as auxiliary targets is largely underexplored in bioacoustics. Gebhard et al. (2024) used sound descriptions, functional traits (Tobias et al., 2022) and life-history (Storchová & Hořák, 2018) as metadata for zero-shot species ID. While these per-species metadata sources give additional context about the species itself, our work explores per-recording metadata that often provide complementary information. NatureLM-audio (Robinson et al., 2024a) and AnimalSpeak (Robinson et al., 2024b) use various recording metadata to construct templated text captions that are used to train audio-language models. In contrast, our work explores each metadata source with a dedicated prediction task, disentangling their effect and enabling more detailed analyses. Chasmai et al. (2026) and Robinson et al. (2024a) also explore the prediction of metadata such as location and vocalization call type as primary tasks for bioacoustic applications beyond species identification. In this work, we focus solely on species identification, comparing its performance with and without metadata.

**Species-only learning.** A large fraction of bioacoustics is devoted to species identification, and methods tend to rely on species labels as the only source of supervision. Prior work has focused on birds (Kahl et al., 2021; Ghani et al., 2023; Tolkova et al., 2021; Cohen et al., 2020; Rauch et al., 2025a; Moummad et al., 2024; Michaud et al., 2023) and broader terrestrial (Chasmai et al., 2024; Miron et al., 2025; Robinson et al., 2024a; Cauzinille et al., 2024; Nolasco et al., 2025; Sarkar & Doss, 2025; Wierucka et al., 2025; Kath et al., 2024; Liu et al., 2026) and marine (Williams et al., 2025; Irfan et al., 2021; Burns et al., 2025) taxonomic groups, leading to foundation models for bioacoustics. Recent work has shown the promise of increasing supervision and including general sound sources (van Merriënboer et al., 2025; Miron et al., 2025). We build on this progress and address the question of whether metadata can be useful sources to provide even more supervised signal and facilitate better transfer under challenging domain shifts.

**General multitask and multimodal learning.** Training with multiple tasks can be challenging because of differing loss and optimization semantics. Prior work has explored dynamic loss weighting (Groenendijk et al., 2021; Kendall et al., 2018), gradient manipulations (Wang & Tsvetkov, 2021; Yu et al., 2020), and multi-objective optimization (Sener & Koltun, 2018) techniques. Strategies for handling missing modalities (Wu et al., 2024) are useful references when handling missing metadata. In this work,

---

[1]Model released at github:google-research/perch/metaperch.

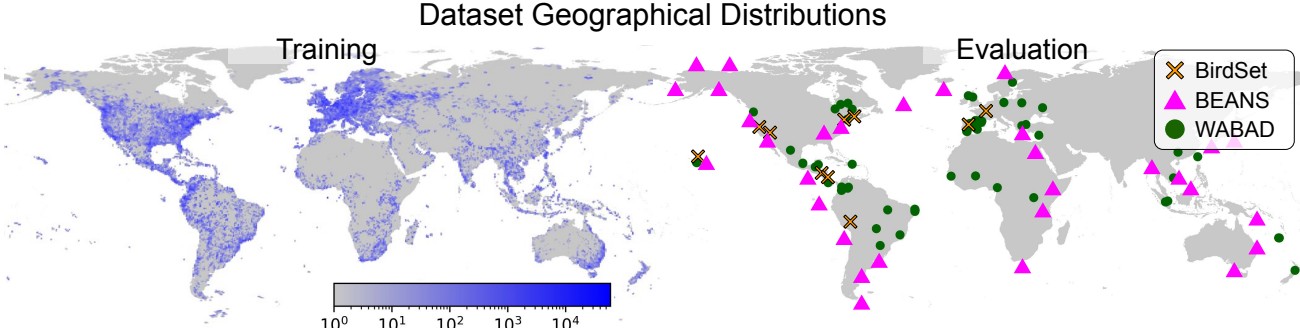

*Figure 2.* **Dataset statistics**. Geographical distribution of the training (log scale) and evaluation (rough locations) datasets.

we address these challenges in the context of bioacoustics and present training recipes for metadata-aware species identification.

**Metadata outside bioacoustics.** The use of metadata, particularly location, has received significant interest in other fields as well. Soundscape mapping (Khanal et al., 2024) aims to understand the acoustic signatures of different geographic locations. Remote sensing approaches also often rely on reconstruction of environmental metadata such as altitude and temperature to learn robust geospatial embeddings (Brown et al., 2025; Klemmer et al., 2025). Similar to bioacoustics, conditioning on location to form geographic priors has been explored in visual species identification (Mac Aodha et al., 2019). Directly predicting metadata such as location has also received recent attention with images (Vivanco Cepeda et al., 2023; Haas et al., 2024) as well as audio (Chasmai et al., 2026). While the use of metadata has seen some success in these fields, it remains under-explored in bioacoustics, and whether the successes transfer to this challenging domain is an open question that we aim to address in this work.

## 3. Methodology

Our primary goal with this work is to investigate the impact of a wide range of metadata sources on model performance when used to construct auxiliary supervision targets. We borrow elements of the base architecture design from the current SotA bioacoustics foundation model, Perch 2.0 (van Merriënboer et al., 2025), and modify key elements to enable faster iteration and better visibility of metadata influence, to establish a strong baseline for comparison which we refer to as BioBaseline. We build upon this baseline architecture and introduce auxiliary metadata prediction tasks. After careful model selection among models co-trained with metadata supervision signals, we compare the best-performing model, called METAPERCH, against BioBaseline and other recent bioacoustics foundation models on a wide range of evaluation tasks.

*Table 1.* **Dataset statistics**. Summary of the number of classes and total size (in hours) for the training datasets.

| Dataset | Type | Classes | Hours |
|---|---|---|---|
| Xeno-Canto | species | 12,308 | 13,555 |
| iNaturalist | species | 8,382 | 3,077 |
| Tierstimmenarchiv | species | 2,375 | 1,462 |
| FSD50K | general | 198 | 80 |
| Total | | 14,795 | 18,174 |

The following subsections describe BioBaseline and METAPERCH in more detail. We focus on the specific design choices behind METAPERCH in this section and present a large-scale study of these considerations and ablations in Section 4.4 and the Appendix.

### 3.1. Training datasets

Following van Merriënboer et al. (2025), we train BioBaseline and METAPERCH on four labeled audio datasets, using the same class mappings and spectrogram configurations: Xeno-Canto (Vellinga & Planqué, 2005), iNaturalist (iNaturalist contributors, 2025), the Tierstimmenarchiv (Frommolt, 1996) and FSD50K (Fonseca et al., 2022). The first three predominantly feature animal vocalizations while the last contains a variety of anthropogenic and environmental sounds. In total, this collection of datasets constitutes over 18,000 hours of audio from 1.55M recordings, with 1.62M annotations across 14,597 animal species and 198 general sound event classes. Refer to Table 1 for summary datasets statistics and Figure 2 for geographical distributions of the training and evaluation datasets. Additional details are included Appendix A.1.

### 3.2. BioBaseline

In terms of architecture and training, we broadly follow van Merriënboer et al. (2025) for BioBaseline. We use random window sampling with 5s windows and *mixup* augmentation (Zhang et al., 2018; van Merriënboer et al., 2025). The

2D spectrogram of this mixed window is treated as a single-channel image and embedded using EfficientNet-B3 (Tan & Le, 2019), a lightweight vision model with ∼12M parameters. Activations of the penultimate layer, before pooling, are extracted as the spatial embeddings. The model includes a prototype learning classifier (Chen et al., 2019; Heinrich et al., 2025) with these spatial embeddings and a linear classifier with spatially-aggregated embeddings to predict the class. Both heads are trained with a cross-entropy loss, but the embedding model receives gradients back-propagated from the linear head only.

Two components we do not include which differentiate BioBaseline and METAPERCH from Perch 2.0 are source prediction and self-distillation. Source prediction—predicting which recording a window was sampled from—requires the embeddings of each window to encode information to identify the original audio recording. We posit that this task may involve implicitly learning information that is encoded in various metadata sources, which can obfuscate the influences of different metadata on the resulting model. Additionally, the output head is the dimension of the number of recordings, which substantially increases the number of model parameters given our training corpus. Self-distillation involves a two-stage training regime—first training a "noisy student" model, then using that to learn a fine-tuned model—where hyperparameter tuning is required at each training stage. This process is computationally expensive and reduces scalability.

## 3.3. METAPERCH

What differentiates METAPERCH from other biacoustics foundation models is the addition of auxiliary losses during training. In conjunction with species detection, the model receives supervised signals by predicting corresponding metadata values from the input audio. We construct additional metadata prediction heads as small multilayer perceptrons (MLPs) each project the spatially-aggregated embedding into a dimension appropriate for the given metadata. The specific architecture details for each MLP—number of hidden layers, hidden dimension, and activation function—are treated as hyperparameters.

From our large-scale empirical study, we find that location, season, and background species as auxiliary prediction tasks yield a strong model. These metadata are also intuitively correlated with species identification. Predicting location incentivizes the model to also attend to ambient and background sounds for potential geographic cues, which should be helpful in soundscapes. As in Weyand et al. (2016), we group neighboring locations into S2 cells (Google, 2025) and treat location prediction as a classification task. In conjunction with location, the season can add useful context for migratory species. A fraction

of the recordings also include annotations for the background species, and attending to these could further facilitate better transfer to soundscapes. For an overview of the methodology, please refer to Figure 3. The other metadata sources considered, but not used after model selection, such as recording characteristics (quality, sampling rate), and species individual attributes (vocalization call type, individual life stage) are also included in Figures 3 and 4, and detailed in Appendix A.4. Our model, and the general multi-task learning paradigm is formalized in Appendix A.10.

### 3.3.1. LEARNING FROM METADATA

**Multi-loss training.** Appropriate loss balancing is important for effective optimization in multi-task learning. We frame most metadata prediction tasks as classification problems and use the (softmax) cross-entropy loss—we find empirically that this provides useful training signal, and also yields similar loss landscapes and orders of magnitude across the tasks. For simplicity, we use a constant loss weighting strategy and include the weight of each metadata loss as a hyperparameter.

**Missing metadata.** While metadata such as location are available for most recordings, others such as background species or recording notes are missing for significant fractions of recordings, some for entire datasets (Figure 4). To fully utilize the metadata that is available, it becomes important to appropriately handle cases of missing metadata. Following prior work in multimodal learning with missing modalitites (Wu et al., 2024), we assign placeholder values for recordings that are missing a particular metadata and then zero-out their loss contributions.

**Mixup with metadata.** Metadata requires more care than species labels when *mixup* is involved. When mixing two recordings with distinct species, it is reasonable to define the target as the union of species present in both recordings. However, defining the location metadata targets as the average of the two recordings' latitude and longitude may result in nonsensical targets, for example. In practice, the metadata prediction tasks used in METAPERCH are all framed as classification tasks, and we found that naively constructing a multihot target with all individual classes worked well.

Another edge case to consider is how to handle metadata that is only available in some but not all of the mixed recordings. In such cases, we still mix the audio from all recordings, but only mix metadata that is present. Although this design leads to potential mislabeling for cases with missing metadata, in practice, we observe that it works better than the other alternatives we explored.

**Adversarial learning.** Some species-metadata correlations may reflect training biases rather than ecologically

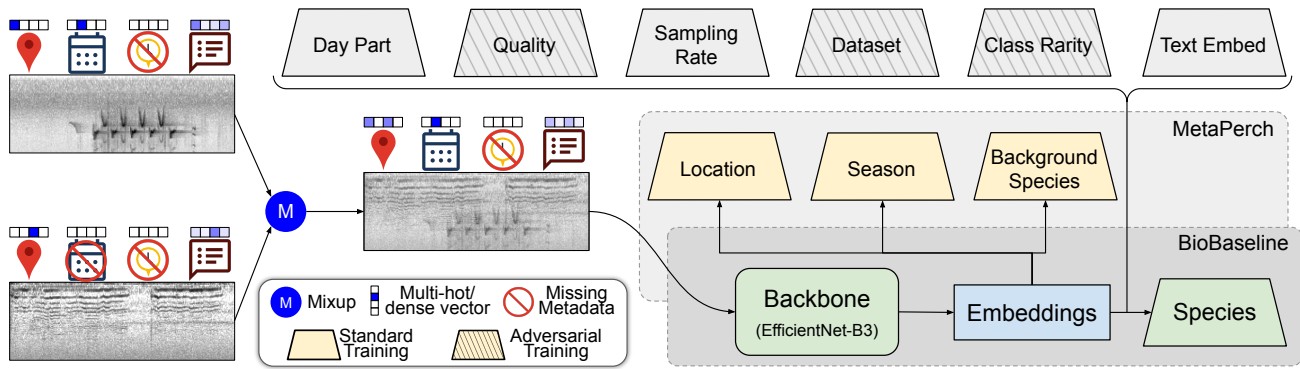

*Figure 3.* **Method diagram**. Overview of BioBaseline and METAPERCH.

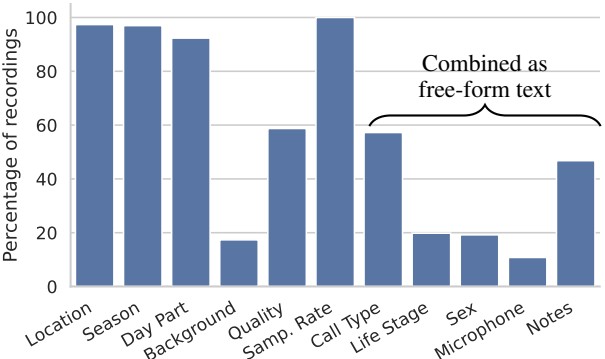

*Figure 4.* **Metadata availability**. Percentage of recordings across the full training dataset which have each metadata type.

grounded relationships. Using such metadata for auxiliary training may actually make the model more brittle to domain shift. In such cases, we can encourage embeddings to be agnostic to—rather than predictive of—such "spurious" metadata by leveraging an idea introduced in domain adversarial networks (Ganin et al., 2016) and using a gradient reversal layer (Osumi et al., 2019) between the backbone and the metadata prediction heads. By maintaining the forward pass while negating the backward pass, this layer ensures that the metadata head is trained with the usual gradients while incentivizing the model to learn embeddings that worsen metadata prediction. Our model selection procedure is allowed to explore both standard and adversarial training for each metadata source.

### 3.4. Validation datasets and model selection

In alignment with the model selection procedure presented in van Merriënboer et al. (2025), we validate on tasks that mirror some aspects of the evaluation setup while ensuring that there is no test set leakage. We perform model selection based on three tasks categories:

1. **observed-species classification** (Powdermill (Denton et al., 2022) and Caples (Denton et al., 2022))
2. **one-shot embedding retrieval** (training sets of BEANS detection tasks, Weldy calltype (Weldy et al., 2024), and Powdermill and Caples)
3. **novel species linear probing** (training sets of BEANS classification tasks, DCLDE (Palmer et al., 2025), NOAA (NOAA Pacific Islands Fisheries Science Center, 2021), ReefSet (Williams et al., 2025), and some other tasks from Ghani et al. (2023))

We average the ROC-AUCs in each validation task category and then compute a geometric average over the three categories, which we use for model selection. To tune hyperparameters, we use Vizier (Golovin et al., 2017), a black-box optimization algorithm. Refer to Appendix A.6 for an overview of our hyperparameter optimization search spaces and selected values.

## 4. Results and discussion

We investigate the impacts of metadata through the performance of METAPERCH on an extensive evaluation suite consisting of 17 species identification datasets. For each benchmark, we describe the evaluation setup, then discuss our performance on it. For hyperparameter tuning, we use Vizier (Golovin et al., 2017), Python-based service for black-box optimization, with identical numbers of trials to perform model selection for both METAPERCH and BioBaseline. The selected hyperparameters are then used to retrain models 5 times with different random seeds to compute the mean and standard deviation on each metric. We report the mean performance in the tables below, and include additional results with standard deviations in Appendix A.7.

### 4.1. BirdSet benchmark

We present our results on BirdSet (Rauch et al., 2025b) in Table 2 and Figure 5. The benchmark includes seven avian datasets from North and South America, the Hawai'ian is-

*Table 2.* **BirdSet benchmark results.** Species identification performance on BirdSet. Our results are means of 5 runs. For each dataset, **bold**: best; underline: second best. Improvements over the BioBaseline baseline are shown in the last row.

| Method | PER | NES | UHH | HSN | NBP | SSW | SNE | Mean | |
|---|---|---|---|---|---|---|---|---|---|
| | ROC-AUC (Trained Classification Head) | | | | | | | ROC-AUC | cmAP |
| Perch 1.0 | 0.700 | 0.900 | 0.760 | 0.860 | 0.910 | 0.910 | 0.830 | 0.839 | 0.356 |
| Audio ProtoPNet-5 | 0.790 | 0.930 | 0.870 | **0.920** | 0.930 | 0.970 | 0.860 | 0.896 | 0.424 |
| BirdMAE-L | **0.820** | 0.910 | 0.820 | 0.900 | **0.940** | 0.930 | 0.880 | 0.886 | **0.440** |
| Rauch et al. (2025b) | 0.720 | 0.880 | 0.790 | 0.890 | 0.920 | 0.930 | 0.830 | 0.851 | 0.360 |
| Perch 2.0 | 0.786 | **0.953** | **0.912** | 0.915 | 0.933 | **0.973** | **0.883** | **0.908** | 0.431 |
| BioBaseline | 0.756 | 0.942 | 0.874 | 0.893 | 0.935 | 0.971 | 0.868 | 0.891 | 0.432 |
| METAPERCH | 0.801 | 0.948 | 0.900 | 0.909 | 0.937 | 0.971 | 0.874 | 0.906 | 0.438 |
| | (+0.045) | (+0.006) | (+0.026) | (+0.016) | (+0.002) | (+0.000) | (+0.006) | (+0.015) | (+0.006) |

*Table 3.* **BEANS benchmark results.** Species identification performance on BEANS. Our results are means of 5 runs. For each dataset, **bold**: best; underline: second best. Improvements over the BioBaseline baseline are shown in the last row. For brevity, we use shorthands for some datasets (ENA: ENABirds, Humbug: HumbugDB, Gibbon: Hainan Gibbons).

| Method | Watkins | Bats | Dogs | Humbug | DCASE | ENA | Hiceas | RFCX | Gibbon | Mean | |
|---|---|---|---|---|---|---|---|---|---|---|---|
| | Classification (Accuracy) | | | | Detection (cmAP) | | | | | Acc | cmAP |
| AVES-Bio | 0.879 | 0.748 | 0.950 | 0.810 | 0.392 | 0.555 | 0.629 | 0.130 | 0.284 | 0.847 | 0.398 |
| Perch 1.0 | 0.855 | 0.718 | 0.942 | 0.739 | 0.283 | 0.603 | 0.502 | **0.232** | 0.146 | 0.814 | 0.353 |
| BirdNet | 0.897 | 0.706 | 0.885 | 0.782 | 0.455 | 0.648 | 0.431 | 0.148 | 0.279 | 0.818 | 0.392 |
| BioLingual (FT) | 0.894 | 0.766 | **0.971** | **0.817** | 0.475 | 0.688 | **0.677** | 0.178 | 0.376 | 0.862 | 0.479 |
| NatureLM-Audio | 0.788 | – | – | 0.114 | 0.058 | 0.314 | 0.336 | 0.025 | 0.005 | – | 0.148 |
| Miron et al. (2025) | **0.914** | 0.681 | 0.906 | 0.789 | 0.465 | 0.566 | 0.527 | 0.118 | 0.366 | 0.823 | 0.408 |
| Perch 2.0 | 0.870 | 0.804 | 0.942 | 0.770 | 0.469 | 0.751 | 0.575 | 0.200 | 0.515 | 0.847 | 0.502 |
| BioBaseline | 0.884 | 0.804 | 0.944 | 0.786 | 0.477 | 0.778 | 0.563 | 0.196 | **0.519** | 0.854 | 0.506 |
| METAPERCH | 0.905 | **0.816** | 0.961 | 0.798 | **0.496** | **0.787** | 0.568 | 0.209 | 0.500 | **0.870** | **0.512** |
| | (+0.021) | (+0.012) | (+0.017) | (+0.012) | (+0.019) | (+0.009) | (+0.005) | (+0.013) | (-0.019) | (+0.016) | (+0.006) |

lands, and Europe. Each PAM recording is accompanied with dense species annotations (bounding boxes), which we treat as a detection problem. Since all species in these datasets are present in our training corpus, we use the trained prototype learning head directly for species prediction and compute ROC-AUC scores.

**Metadata improves transfer to soundscapes for known species.** BirdSet captures model performance under the acoustic (focal → soundscape) distribution shift. We see consistent improvements for METAPERCH over BioBaseline across datasets, with an increase of 0.015 in ROC-AUC on average. This improvement could be partly attributed to learning embeddings that attend to and are predictive of background species. METAPERCH also performs favorably in comparison to recent approaches: its ROC-AUC is 0.002 below the SotA, Perch 2.0, a more complex model.

**Metadata helps more in underrepresented regions.** The geographic distribution of our training datasets (Figure 2) is highly imbalanced: South America and Hawai'i, among

others, are underrepresented compared to North America and Western Europe. In BirdSet, we observe strong performance for PER (Peru) (Hopping et al., 2022), UHH (Hawai'i) (Navine et al., 2022), and NES (Colombia, Costa Rica), with an improvement of 0.025 ROC-AUC points on average. On the other hand, the North American and Western European datasets (HSN (Clapp et al., 2023), NBP (Morfi et al., 2019), SSW (Kahl et al., 2022a), SNE (Kahl et al., 2022b)) have relatively smaller improvements, averaging at 0.006 ROC-AUC points. These results suggest that metadata auxiliary training signal—particularly from location and season—encodes additional information that may help bridge the data gap for underrepresented regions.

### 4.2. BEANS benchmark

The BEnchmark of Animal Sounds, BEANS (Hagiwara et al., 2023), includes 12 bioacoustic and miscellaneous audio datasets. For each dataset, a train split is included, which we use to train prototype learning probes on the

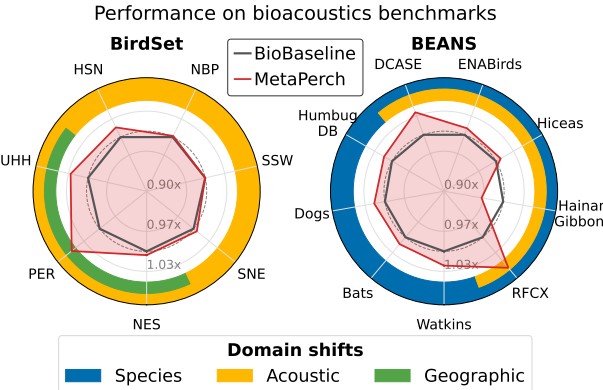

*Figure 5.* **Performance on bioacoustic benchmarks.** Visualization of our performance relative to the baseline. For each dataset, we present the same metric used in Tables 2 and 3 (ROC-AUC for BirdSet, Accuracy & cmAP for BEANS). We highlight the domain shifts (species, acoustic, geographic) in each dataset to better illustrate scenarios where metadata is beneficial.

model's frozen embeddings. We treat datasets with focal recordings as classification tasks and report accuracy, and those consisting of PAM soundscapes (with dense labels) as detection tasks and report cmAP. Our results are presented in Table 3 and Figure 5. Among the datasets presented in BEANS, we include only the bioacoustic tasks, and exclude CBI (Howard et al., 2020) because of potential overlap with our training set.

**Metadata helps transfer to novel species for focal recordings.** BEANS classification tasks include focal recordings with species and classes either completely unobserved or relatively rare in our training corpus. We observe consistent benefit on all classification tasks, with an improvement of 0.016 on average accuracy. Notably, on Watkins (Sayigh et al., 2016) and Bats (Prat et al., 2017) accuracy improves by 0.021 and 0.012, respectively, even though aquatic and very high frequency bat sounds differ greatly from the predominantly avian sounds in our training data. HumbugDB (Kiskin et al., 2021) contains mosquito data from different parts of the world, and encoded location information may explain METAPERCH's better performance. METAPERCH also seems to better capture intraspecies variations in the Dogs dataset (Yin & McCowan, 2004).

**Metadata helps transfer to soundscapes for novel species.** Performance on the BEANS detection tasks captures model performance under both acoustic and species shift. METAPERCH also generally outperforms BioBaseline here, with an improvement of 0.006 on Mean cmAP. We observe strong improvements of 0.009 and 0.019 ROC-AUC on the avian datasets ENABirds (Chronister et al., 2021) and DCASE (Morfi et al., 2021), respectively. In contrast, metadata has limited benefit for the other datasets

Hiceas (NOAA Pacific Islands Fisheries Science Center, 2022), RFCX (LeBien et al., 2020), and Hainan Gibbons (Dufuorq et al., 2021), which feature marine mammals, anurans, and primates, respectively. We suspect that while METAPERCH improves transfer under species and acoustic shifts on their own, the benefits diminish when both shifts are present in conjunction.

Overall, METAPERCH achieves SotA performance on BEANS classification and detection tasks, outperforming the next best model by 0.008 accuracy and 0.006 cmAP points.

### 4.3. WABAD

WABAD (Pérez-Granados et al., 2025) is a soundscape dataset consisting of 1,192 bird species from around the world. It is geographically diverse, particularly with recordings from Africa, Asia, and New Zealand—regions that are less prominent in the other evaluation datasets. Similar to BirdSet, we use the trained classifier heads (linear and prototype learning) directly to compute ROC-AUC. We also report a one-shot retrieval metric, similar to the validation tasks. Our results are presented in Table 4.

**Metadata helps in globally distributed tasks.** While the other benchmarks include datasets from different parts of the world, most tasks are geographically limited on their own, and the model is tasked with differentiating recordings within a task only. For tasks with species from different parts of the world, correlations with metadata such as location can be incredibly useful. In WABAD, META-PERCH improves over the baseline by 0.070 and 0.018 (ROC-AUC) for linear and prototype learning probes, respectively. Table 3 shows similar results with the other globally distributed tasks in BEANS: Watkins and HumbugDB.

**Metadata benefits vary across biomes.** We evaluate the performance of models on recordings from different biomes in WABAD. The Tropical and Sub-tropical biome has the highest improvements of 0.023, while for deserts, we see a marked drop of 0.024 ROC-AUC. These results suggest that the benefits for metadata are not uniform, and that recording environment may be correlated to the benefit of metadata for species identification.

**Metadata allows embeddings to encode habitat information.** In Figure 6, we also visualize our and baseline embeddings, colored by the biomes. Our embeddings appear more separable by biome, albeit still fragmented, with distinct sub-clusters forming within each biome. This structure is also visible in the visualizations colored by species in Appendix A.8. Since this is a multi-species detection dataset, we suspect that these sub-clusters correspond to species combinations that co-occur frequently.

*Table 4.* **WABAD results.** Species identification performance on WABAD, over the full dataset (left), and separated by biome (right). Our results are means of 5 runs. For each metric, **bold** represents the best performance.

| Method | Mean ROC-AUC | | |
| --- | --- | --- | --- |
| | 1-shot | Linear | Proto |
| Perch 2.0 | **0.919** | 0.739 | 0.924 |
| BioBaseline | 0.912 | 0.741 | 0.928 |
| METAPERCH (ours) | 0.917 | **0.811** | **0.946** |

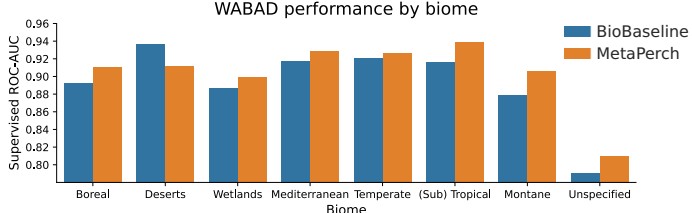

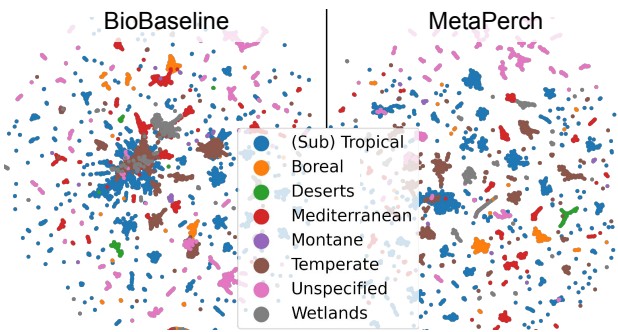

*Figure 6.* **Embedding visualizations**. 2D UMAP projections of BioBaseline (left) and METAPERCH (right) embeddings of WABAD windows, colored by biome.

| Method | WABAD | Val | Method | WABAD | Val |
| --- | --- | --- | --- | --- | --- |
| Best Vizier | 0.946 | **0.880** | METAPERCH | 0.946 | **0.880** |
| Median Vizier | **0.952** | 0.870 | Mix iff all | 0.945 | 0.874 |
| Gradient Surgery | 0.944 | 0.874 | Weighted mean | **0.947** | 0.876 |
| Dynamic equalize | 0.905 | 0.875 | No *mixup* | 0.941 | 0.869 |
| *(a)* Multi-loss training | | | *(b)* Metadata *mixup* variants | | |

*Table 5.* **Ablations**. We report ROC-AUC of prototype learning head on WABAD as well as the model-selection validation metric.

## 4.4. Ablations

**METAPERCH design choices.** First, we study the effects of different design choices in METAPERCH. For the multi-loss training strategy (Table 5a), our validation performance suggests that treating the weights of each metadata loss as hyperparameters to be optimized works better than more involved strategies such as dynamic equalization or gradient surgery (Yu et al., 2020). This is likely because all of our losses are softmax cross entropies, have similar gradient landscapes and are in similar orders of magnitudes. For *mixup* (Table 5b), we observe that our mixing of any present metadata attains better validation performance than the stricter choice of mixing a metadata if and only if (iff) it is present for all. Using a weighted mean also leads to similar drops, and all 3 are better than no *mixup* at all.

In these experiments, we note a disconnect between the validation and WABAD performance: for example, after sorting the Vizier trials by their validation performance, the median trial actually performs better on WABAD than the best one. Since our design choices are driven by the validation performance, we may miss out on some test performance improvements, and these results highlight the challenges of choosing appropriate validation tasks.

**BIRB.** We analyze the performance of METAPERCH under controlled distribution shift experiments, similar to those proposed in BIRB (Hamer et al., 2023). Focusing on the Hawaiʻi (Navine et al., 2022) and Coffee Farms (Álvaro

Vega-Hidalgo et al., 2023) datasets from BirdSet, we exclude all species present in these datasets from our training data and retrain BioBaseline and METAPERCH. The task is framed as few shot classification, where a few iNaturalist windows of each species are used to train a linear probe, which is then evaluated on 1) the Xeno-Canto subset containing held-out species and 2) the PAM datasets themselves. While the latter contains both species distribution and acoustic shifts, the former allows us to study species distribution shift in isolation.

These results are presented in Figure 7a. For lower shots, we observe that METAPERCH outperforms BioBaseline for both Xeno-Canto held-out and the soundscape datasets. This trend reverses for higher shots in the Xeno-Canto held-out set, suggesting potential shortcomings of METAPERCH. The Xeno-Canto held-out set reflects an artificial setting designed to disentangle the effects of acoustic and species shifts, whereas PAM soundscape experiments are more representative of natural deployment conditions. We report on both to provide a holistic picture of METAPERCH, but we expect PAM soundscape results to be more reflective of actual metadata benefits in practice.

**Effects of missing metadata.** We expect the benefits offered by a particular metadata to increase as more of this metadata is available. To better understand the sensitivity of model performance to metadata availability, we observe performance with location prediction when fractions of the metadata are artificially marked as missing. The results, presented in Figure 7b, show that while the performance decreases as lesser metadata is present, there are still noticeable improvements over baseline for as low as 1% availability. This further highlights the strong potential of metadata, and supports our choice of including metadata even if

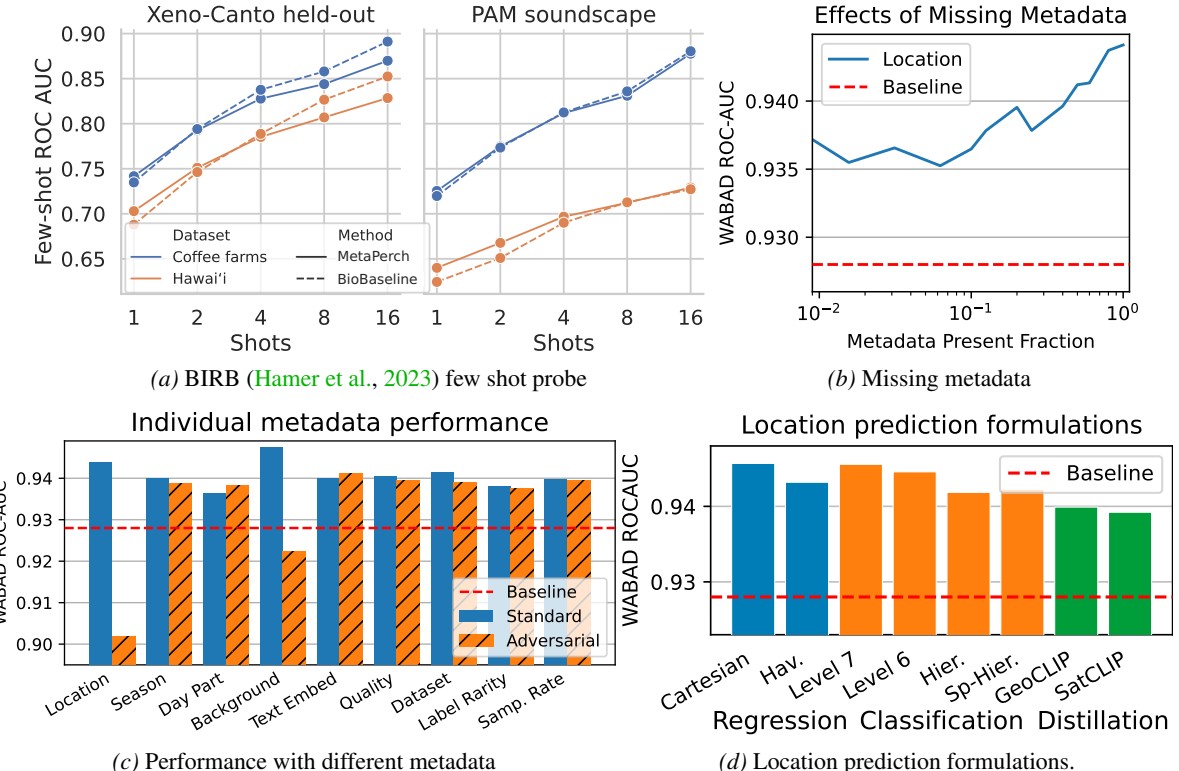

*(a)* BIRB (Hamer et al., 2023) few shot probe

*(b)* Missing metadata

*(c)* Performance with different metadata

*(d)* Location prediction formulations.

*Figure 7.* **Ablations**. For (a), we report few-shot linear probe performance on BIRB (Hamer et al., 2023). For (b-d), we report ROC-AUC achieved by prototype learning head on WABAD.

they are available for only a fraction of the training data.

**Individual metadata performance.** We observe improvements by encouraging features to be predictive of most metadata on their own, with boosts ranging from 0.008 for Day-Part to 0.019 ROC-AUC points for Background Species (Figure 7c). Adversarial training is better for day-part and text embeddings. Interestingly, we observe similar performance for both standard and adversarial training with sampling rate and quality. Background species and location have the most improvement with standard training, and the most drops with adversarial training, highlighting their importance for robust bioacoustic features.

**Location prediction formulations.** Each metadata prediction task can be formulated in a number of ways, with corresponding prediction heads and losses. We explore some formulations of location prediction in Figure 7d (additional details in Appendix A.4). Strong WABAD performance across different formulations (regression, classification, and distillation) underscores the potential of location prediction. S2 cell classification at level 7 (average cell area is roughly 50,000km$^2$) performs best, followed closely by Cartesian regression. Finding a suitable loss formulation involves a tradeoff between learnability and specificity. For example, a coarser-grained S2 cell mapping may be easier to predict, but misses finer geographical context.

## 5. Conclusion and future work

Metadata offers a rich source of auxiliary supervision for bioacoustics foundation models: learning with metadata improves species identification by leveraging species–metadata correlation. Global patterns in readily available metadata such as location and time can help species identification and increase robustness to various domain shifts. Our study demonstrates the potential of using such metadata as auxiliary losses, highlights the most beneficial metadata sources, and explores the effects of different design choices in this paradigm. Strong performance benefits with our simple approach presents a lower bound for metadata benefits and lays the groundwork for future research in metadata-aware species identification.

An interesting future direction is in the design of alternative paradigms such as pre-training or test-time conditioning. Extending our method to other relevant metadata and base foundation models would make for meaningful follow-up work. Metadata could also guide data sampling in an effort to alleviate some spatiotemporal biases or construct training curricula. We hope the evidence presented in this work motivates researchers to consider incorporating diverse metadata and auxiliary supervision signals when constructing datasets and benchmarks.

## Acknowledgements

We thank Bart van Merrienboer, Tom Denton and Lauren Harrell for their perspectives around this work. We thank Richard Song for guidance on deploying large-scale Vizier hyperparameter tuning.

## Impact statement

**Potential misuse: wildlife disruption or harm.** An important goal of bioacoustics research in general is to aid conservation efforts and biodiversity monitoring by automating and scaling species identification. However, these models could also be used to harm species of interest. For example, species of interest could be identified using bioacoustic monitoring setups with species identification to locate species of interest, and this information could be used to harass, trap, or poach species.

For METAPERCH in particular, metadata such as location can be another cause for concern. While we do not release metadata prediction heads, our embeddings may still be used to train geolocalization models to locate species of interest. However, the S2 cell level used (L7) is coarse enough (average cell area of around 50,000 square kilometers) that predicting the S2 cell at that level would not be useful for poaching purposes.

**Personally identifiable information.** Human voice or other personally identifiable content may be present in the citizen science platforms such as iNaturalist and Xeno-Canto, and consequently, our training data. Since the recordings are openly available and licensed for research use, we do not obfuscate or modify the data for training.

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

# A. Appendix

## A.1. Additional dataset details

We enumerate all datasets used in our experiments in Table A1. The species in the different training datasets partially overlap. In total, we have more than 18,000 hours of audio from 1.55M recordings with 1.62M annotations across 14,597 species and 198 general sound event classes. Following van Merriënboer et al. (2025), the Xeno-Canto data was obtained from the public API and the iNaturalist data was obtained by collecting the 'research-grade' audio examples reported to the Global Biodiversity Information Facility (GBIF). Since Xeno-Canto, iNaturalist and the Tierstimmenarchiv use different taxonomies (species names) the classes from Xeno-Canto and Tierstimmenarchiv were mapped into the iNaturalist classes. All bat recordings are removed from the datasets.

We visualize the geographical distributions of individual training datasets in Figure A1, and their class frequency distributions in Figure A2. These demonstrate both the geographical and ecological skew in the training datasets. The long-tailed distributions make species identification particularly challenging for underrepresented regions and rare species. We include addition statistics of metadata availability in Figure A3.

## A.2. Additional metric details

Let the labels for an evaluation task of $N$ audio windows across $C$ species be $y \in \mathbb{R}^{N \times C}$ and the predictions made be $\hat{y} \in \mathbb{R}^{N \times C}$.

**ROC-AUC.** This metric computes the probability of ranking a random positive above a random negative. It captures a model's ability to distinguish between classes in a threshold-independent manner. ROC-AUC is computed for each class in a one-vs-rest fashion, and then averaged over all classes.

$$\mathrm{ROC} - \mathrm{AUC} = \frac{1}{C} \sum_{c=1}^{C} \mathbb{P}\big(\hat{y}_{i,c} > \hat{y}_{j,c} \mid y_{i,c} = 1, \ y_{j,c} = 0\big) \tag{1}$$

**Accuracy.** We report accuracy for BEANS classification tasks, which are single-label window-level prediction tasks on focal recordings. This is a simple top-1 accuracy, and may be brittle for class-imbalanced data in general.

$$\mathrm{ACC} = \frac{1}{N} \sum_{i=1}^{N} \sum_{c=1}^{C} \mathbf{1}\big\{\hat{y}_{i,c} = y_{i,c}\big\} \tag{2}$$

**cmAP.** We report cmAP for BirdSet and for BEANS detection tasks. This metric measures the ranking ability of a model. Average precision is computed for each class in a one-vs-rest fashion, and then averaged over all classes.

$$\mathrm{cmAP} = \frac{1}{C} \sum_{c=1}^{C} \mathrm{AP}_c \tag{3}$$

## A.3. Additional baseline details

**Preprocessing and augmentation.** For each training datum, we sample a group of 5 second windows at random from distinct training recordings, also sampled at random. Since annotations do not include time bounds, we assign all labels of a recording to each of its windows. The audio windows are mixed (Zhang et al., 2018; van Merriënboer et al., 2025) with each other and converted into a spectrogram via STFT, visualizing the frequency ($F$) composition over time ($T$). The number of windows mixed in each datum and the weights used while mixing audio are themselves sampled from beta-binomial and symmetric Dirichlet distributions, respectively. An aggregate of the labels of each original window is used as the label for this mixed window.

**Modeling.** The 2D spectrogram ($T \times F$) is treated as an single-channel image and embedded using EfficientNet-B3 (Tan & Le, 2019), a lightweight vision model with ~12M parameters. Activations of the penultimate layer, before pooling, are extracted as the spatial embeddings ($T' \times F' \times dim$). We use a prototype learning classifier (Chen et al., 2019; Heinrich et al., 2025) with these spatial embeddings and a linear classifier with aggregate embeddings ($dim$) to predict the species.

**Learning.** Multihot class target vectors are scaled to sum to one and used as soft labels in a softmax cross-entropy loss to train the prototype learning and linear classifiers. For the prototype learning head, we include an orthogonality loss (Donnelly et al., 2022; Heinrich et al., 2025) that maximizes prototype diversity. The embedding model receives gradients backpropagated from the linear classifier, but not the prototype learning classifier. This design promotes linear separability of learned features while retaining benefits of the stronger prototypical classifier. The model is trained with an Adam (Kingma & Ba, 2015) optimizer for 500K training steps with a batch size of 256. Please refer to Appendix A.6 for other hyperparameters.

## A.4. Additional method details

**Location.** Species are inherently tied to their geographic ranges and knowing the location of a recording can be informative of the type of species one could expect in it. Predicting location may incentivize our models to also attend to ambient and background sounds for potential geographic cues, which may be helpful in soundscapes. As in Weyand et al. (2016), we group neighboring locations into S2 cells (Google, 2025) and treat location prediction

*Table A1.* **Dataset statistics.** Statistics for the different training, validation, and evaluation datasets.

| Dataset | | Classes | Size (hours) |
|---|---|---|---|
| **Training** | | | |
| Xeno-Canto (Xeno Canto) | | 12308 | 13555 |
| iNaturalist (iNaturalist) | | 8382 | 3077 |
| Tierstimmenarchiv (Frommolt, 1996) | | 2375 | 1462 |
| FSD50K (Fonseca et al., 2022) | | 200 | 80 |
| **Validation** | | | |
| Powdermill (Denton et al., 2022) | | | |
| Caples (Denton et al., 2022) | | | |
| BEANS training splits (Hagiwara et al., 2023) | | | |
| Weldy Call Type (Weldy et al., 2024) | | | |
| DCLDE (Palmer et al., 2025) | | | |
| NOAA (NOAA Pacific Islands Fisheries Science Center, 2021) | | | |
| ReefSet (Williams et al., 2025) | | | |
| Evaluation datasets from Ghani et al. (2023) | | | |
| **Evaluation** | | | |
| BirdSet | PER (Hopping et al., 2022) | 132 | 21 |
| | NES (Álvaro Vega-Hidalgo et al., 2023) | 89 | 34 |
| | UHH (Navine et al., 2022) | 25 | 50 |
| | HSN (Clapp et al., 2023) | 21 | 16 |
| | NBP (Morfi et al., 2019) | 51 | 0.8 |
| | SSW (Kahl et al., 2022a) | 81 | 285 |
| | SNE (Kahl et al., 2022b) | 56 | 33 |
| BEANS-classification | Watkins (Sayigh et al., 2016) | 31 | 1.1 |
| | Bats (Prat et al., 2017) | 10 | 1 |
| | Dogs (Yin & McCowan, 2004) | 10 | 0.5 |
| | HumbugDB (Kiskin et al., 2021) | 14 | 6.7 |
| BEANS-detection | DCASE (Morfi et al., 2021) | 20 | 3.9 |
| | ENABirds (Chronister et al., 2021) | 34 | 1.3 |
| | Hiceas (NOAA Pacific Islands Fisheries Science Center, 2022) | 1 | 2.2 |
| | RFCX (LeBien et al., 2020) | 24 | 15.7 |
| | Gibbons (Dufuorq et al., 2021) | 3 | 10.6 |
| WABAD (Pérez-Granados et al., 2025) | | 1192 | 84 |

as a classification task. The S2 cells can be constructed at different resolutions and the number of classes changes accordingly. We also explore other regression and distillation based approaches in Section 4.4. We explore the following different loss formulations and prediction heads for location:

1. **Regression.** Perhaps the most intuitive approach is to regress the location coordinates directly. We use either a simple L2 loss between the spherical coordinates, their projections on a 3D Cartesian sphere, or the Haversine distance as the loss.

2. **Classification.** Similar to Weyand et al. (2016), we group neighboring locations into S2 cells[2] and pre-

---

[2]https://code.google.com/archive/p/s2-geometry-library/

dict the cell a particular recording would belong to. The S2 cells can be constructed at different resolutions (levels): lower resolution cells would likely make the learning task easier while higher resolution cells can allow learning finer geospatial patterns. Our training dataset has significant geographic skew and the resulting S2 class distribution is long-tailed. To remedy this, we also explore hierarchical S2 cells where higher resolution cells are used for regions with higher density of recordings (or species) and vice versa. We visualize the S2 cells for different levels and hierarchical S2 cells in Figure A4 and Figure A5, respectively.

3. **Distillation.** Following prior work in geolocalization (Vivanco Cepeda et al., 2023; Chasmai et al., 2026), we also explore the use of dense loca-

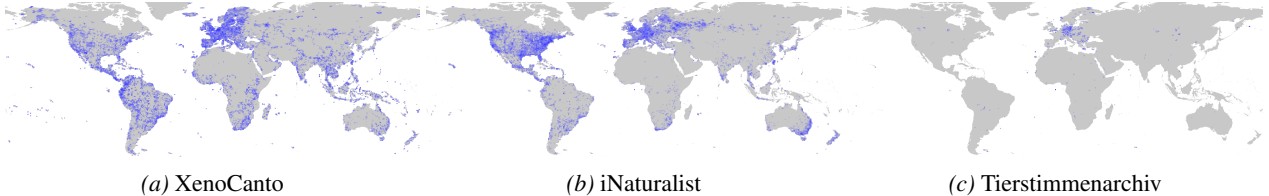

*(a)* XenoCanto      *(b)* iNaturalist      *(c)* Tierstimmenarchiv

*Figure A1.* **Geographical distribution of data in each training dataset.** This demonstrates a high-level view into the geographical skew within and across the datasets.

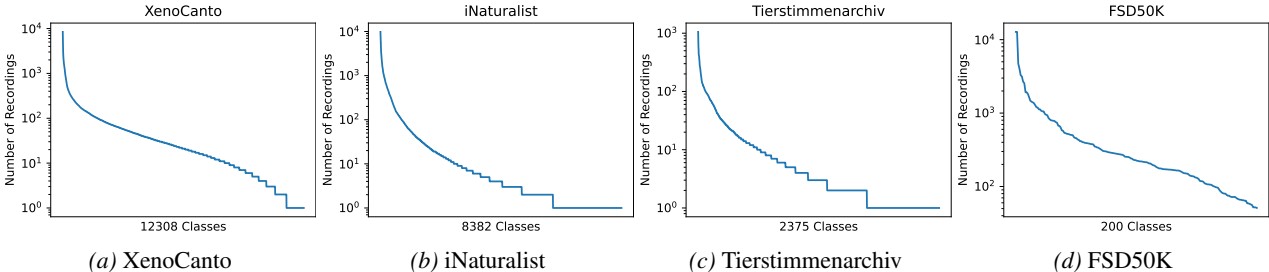

*(a)* XenoCanto      *(b)* iNaturalist      *(c)* Tierstimmenarchiv      *(d)* FSD50K

*Figure A2.* **Species (or other classes) frequency across the training datasets.** Some species (or in the case of FSD50K, other audio classes) are very prevalent in the dataset, while others are much less represented. These plots reflect the nontrivial class distribution skew across each dataset.

tion embeddings as distillation targets. Approaches like GeoCLIP (Vivanco Cepeda et al., 2023) and Sat-CLIP (Klemmer et al., 2025) learn location embeddings with the help of paired ground-level or satellite images. We distill that information into the model by learning to predict the corresponding location embeddings from the model's embeddings (trained on bioacoustic data as inputs only). We use the Cosine similarity between predicted and true location embeddings as the training loss.

**Season.** Species' vocal activity often varies according to the season. In conjunction with location, season can add useful context for migratory species. We use the date and location (specifically the hemisphere) of a recording to determine the season as one of Spring, Summer, Fall or Winter and frame this as a classification task.

**Part of day.** The local time of day could potentially provide a prior over the expected species. For example, audio recorded around dawn is likely to capture a chorus of birds (Gil & Llusia, 2020), while another recorded after dusk may feature more insects. To account for global time zone differences, we use broad boundaries to determine the part of day as one of dawn, dusk, day, and night. This task is also treated as a classification problem.

We explore the following different loss formulations for learning with season or day part:

1. **Classification.** Each of the season or day-part categories are treated as independent classes and the model is trained with cross-entropy.

2. **Unit circle.** Both season and day-part have cyclic temporal relationships between the different categories. To better capture these, we represent the category as a point on the unit circle and task the model to predict the angle or cartesian coordinates. The full circle is divided into equal sectors for each category, so that the last category is followed by the first, capturing the cyclicity of season and day-part. The loss is now the angle between the predicted and true points.

**Background species.** Our training data consists of primarily focal recordings, each accompanied with annotations for the foreground species. A fraction of the recordings also include annotations for the background species, and attending to these could facilitate better transfer to soundscapes. Absence of background annotations does not imply absence of background species. Incorporating them directly into the main prediction task would incorrectly assume otherwise, so we treat background species prediction as metadata and handle missing cases accordingly. We use a loss similar to foreground species prediction.

**Audio characteristics.** Some recordings have additional fields where recordists can include audio quality as categorical grades (A-E or A-C). While this is somewhat subjective, it often captures some measure of the signal-to-noise ratio of the recording. The original sampling rate could also be a useful signal that could allow the model to better address any spectrogram artifacts introduced by resampling. We use the original sampling rate to assign a binary class of whether the recording was up or down-sampled during preprocessing (all resampled to 32kHz).

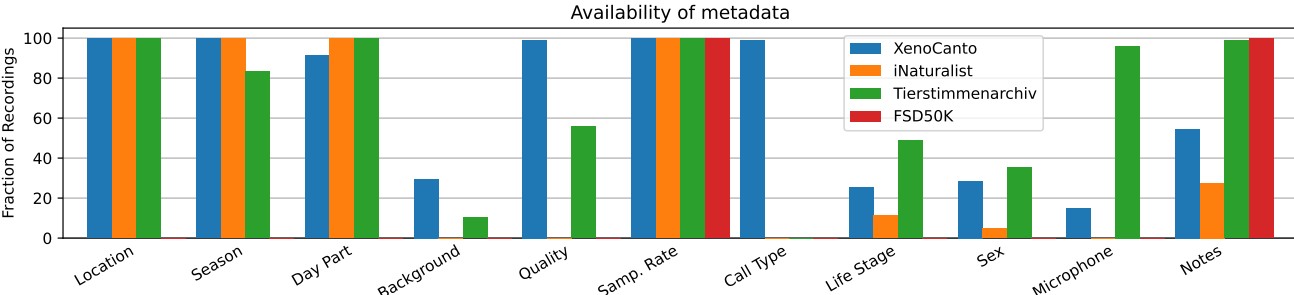

*Figure A3.* **Metadata availability per dataset.** Metadata coverage varies across training datasets.

**Miscellaneous.** Some recordings have the stage of life (juvenile, adult etc) and gender of the species being recorded. FSD50K recordings have textual description and tags while a number of recordings on the bioacoustic platforms have notes left by the recordist. There are also more free-form metadata that hint at different characteristics of the audio such as the microphone type and recording method (field recording, in enclosure etc.). We combine these metadata sources together with the actual foreground label to construct textual prompts and use their text embeddings from the Gemini API as the supervisory targets. We use the negative cosine similarity for training.

**Derived metadata.** We also explore a few artificially constructed metadata. Based on the training species distribution, we categorize each species based on its abundance or rarity and use this as a classification target. We also explore the source of a recording (Xeno-Canto vs iNaturalist, for example) as another metadata.

### A.5. Spectrogram construction

Each audio recording is resampled to 32kHz. We take in audio windows of length 5s (160,000 samples) and construct a log mel-spectrogram using STFT with a hop and window length of 10 and 20ms respectively, outputting a total of 500 frames with 128 mel-scaled frequency bins ranging from 60 Hz to 16 kHz. We use an uncentered Hann window with FFT size of 1024 samples. We calculate the energy (magnitude) spectrogram (so not a power spectrogram) and the output is scaled by the reciprocal of the sum of the window values. After the calculation of the mel-spectrogram we apply a logarithm with a floor of $10^{-5}$ and then multiply the output by 0.1.

### A.6. Hyperparameters

We enumerate the hyperparameters for META​PERCH in Appendix A.6. For each, we mention the search space, defined by the minimum and maximum values in case of continuous parameters and the set of possible values in the case of categorical parameters. For the former, we also include the type of steps taken within the search space (one of linear or logarithmic). Finally, we list the optimal value of each hyperparameter found by Vizier (Golovin et al., 2017).

### A.7. Additional results: BEANS, BirdSet, & WABAD

We include the mean and standard deviations for the 5 runs of META​PERCH and BioBaseline for BEANS (Tables A4 and A5), BirdSet (Table A6), and WABAD (Tables A7 and A8). For BEANS, we also include other datasets that we excluded from the main tables. These datasets were ESC-50, Speech Cmds, and CBI. We also see an improvement over baseline on the average of the full set of datasets, but the ones included in the main paper are more appropriate to our explorations.

### A.8. WABAD additional embedding visualizations

We visualize the embeddings of WABAD, colored by location in Figure A6 and by species in Figure A8. In Figure A9, we visualize the embeddings colored semantically (similar colors indicate similar features) on the world map. Each coordinate is colored by the mean embeddings of all recording windows from the location closest to that coordinate.

### A.9. Additional ablation discussions

**Multi-loss training.** We explore the effects of the multi-loss training strategy in Table 5a. Simply treating the weights of each metadata loss as a hyperparameter to be optimized works better than more involved strategies such as dynamic equalization or gradient surgery (Yu et al., 2020). The likely reason is that since most of our losses are softmax cross entropies, they have similar gradient landscapes and are in similar orders of magnitudes, making the loss weighing problem easier. This is further supported by the relatively small difference in the best and median validation performances we observe during our vizier hyperparameter search. Interestingly, hyperparameters with the median validation score actually performs better on WABAD than the best validation score, highlighting a potential gap between our validation tasks and the evaluation setup.

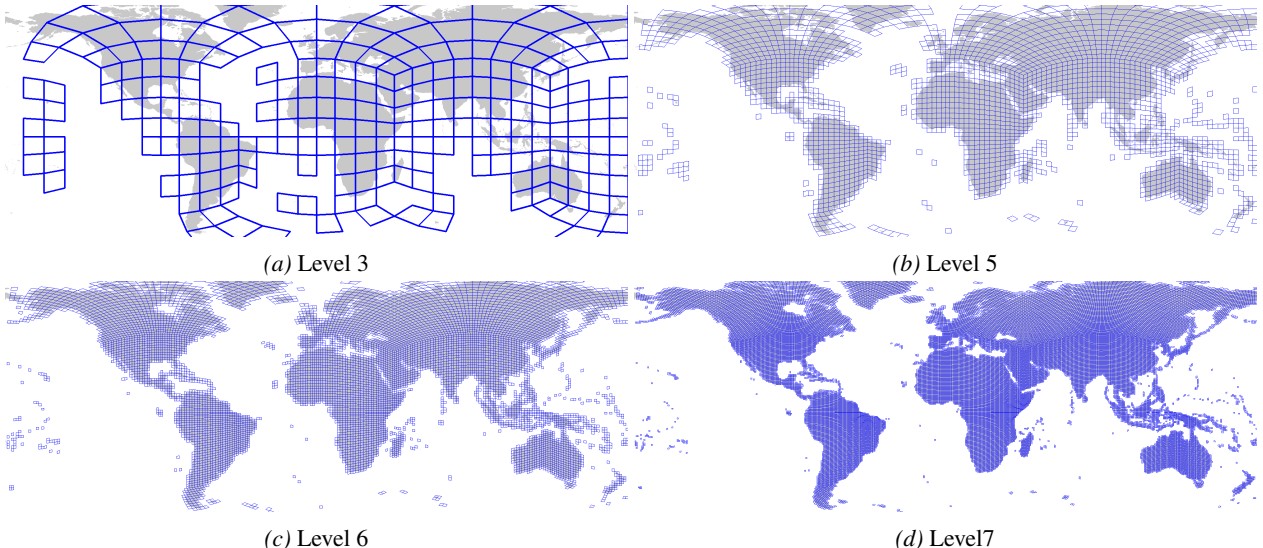

*(a)* Level 3      *(b)* Level 5

*(c)* Level 6      *(d)* Level7

*Figure A4.* **S2 cells.** S2 cell coverage over landmass at different levels. The lower the level, the lower the granularity of each cell.

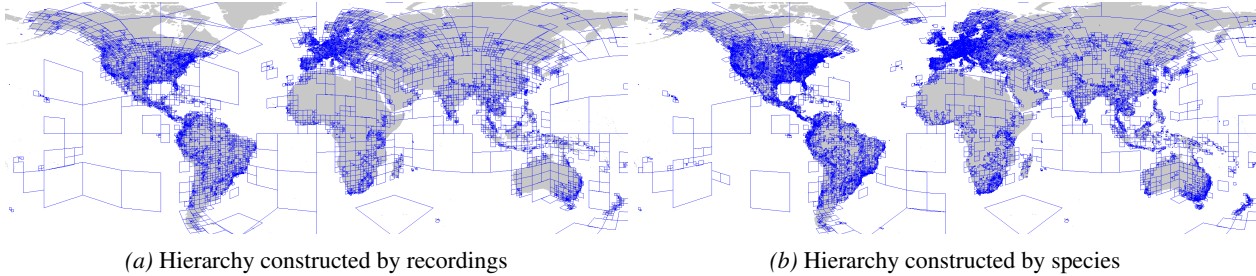

*(a)* Hierarchy constructed by recordings      *(b)* Hierarchy constructed by species

*Figure A5.* **Hierarchical S2 cells.** S2 cell coverage over landmass at levels corresponding to the number of different species present, such that each cell level is thresholded to create a uniform concentration of coverage for each cell across the spatial mapping.

**Mixup variants.** In METAPERCH, we mix any metadata present for a recording. We could be more strict, and mark a mixed metadata present if and only if (iff) it is present for all original recordings. This leads to a drop in validation performance (Table 5b), suggesting that noisy-but-more metadata supervision may be better then clean-but-less. Using the sampled audio-*mixup* weights to get a weighted mean of the metadata instead of a simple mean leads to a small drop. We observe marginal differences in WABAD performance, suggesting it is less sensitive to this design choice. We see significant drops in performance if we remove *mixup* altogether.

**Location loss formulations.** Each metadata prediction task can be formulated in a number of ways. We explore some formulations of location prediction in Figure 7d. Finding a suitable loss formulation involves a tradeoff between learnability and specificity. For example, coarser S2 cells may be easier to predict, but would likely miss fine-grained geographical context. However, we observe strong WABAD performance across different formulations: regression, classification, and distillation. This is likely an

*Table A2.* **Earth embedding models.** Ablation on the use of AlphaEarth (Brown et al., 2025), a strong Earth embedding model, as an alternative formulation of the location prediction task.

| Model | WABAD |
|---|---|
| BioBaseline | 0.928 |
| BioBaseline + classification (level 7) | 0.946 |
| BioBaseline + AlphaEarth | 0.942 |

artifact of these formulations capturing the same underlying information, and further underscores the potential benefits of location prediction.

Following a similar procedure as our GeoCLIP and SatCLIP experiments, we use AlphaEarth (Brown et al., 2025) embeddings corresponding to the location and time (year) of our recordings as distillation targets and train our decoder with cosine loss. Results on WABAD are reported in Table A2. While the inclusion of AlphaEarth improves performance over the baseline, it falls short of classification with level 7, the formulation used in METAPERCH.

*Table A3.* **Hyperparameters**. Search spaces and selected values.

| Hyperparameter | Optimal | Search Space | Steps |
|---|---|---|---|
| **Optimization** | | | |
| Learning Rate | 0.00029784742888791223 | (1e-5, 1e-2) | Logarithmic |
| Dropout | 0.38900657468779043 | (0.1, 1.0) | Linear |
| *mixup* | | | |
| Metadata mix if any present | True | {True, False} | Categorical |
| Dirichlet Concentration | 5.7296725458626 | (1, 100) | Logarithmic |
| Maximum signals to mix | 5 | {2, 3, 4, 5} | Categorical |
| beta_binomial_0 | 15.085589497260358 | (1, 100) | Logarithmic |
| beta_binomial_1 | 12.839225337610824 | (1, 100) | Logarithmic |
| **Background Species** | | | |
| Loss Weight | 1.0419281708242019 | (0, 1.5) | Linear |
| Prediction head activation | relu | {"relu", "swish", "gelu"} | Categorical |
| Prediction head hidden dimension | 512 | {64, 128, 265, 512} | Categorical |
| Prediction head hidden layers | 1 | {0, 1, 2, 3} | Categorical |
| Adversarial Training | False | {True, False} | Categorical |
| **Location** | | | |
| Loss Weight | 1.0062746756429837 | (0, 1.5) | Linear |
| Prediction head activation | swish | {"relu", "swish", "gelu"} | Categorical |
| Prediction head hidden dimension | 265 | {64, 128, 265, 512} | Categorical |
| Prediction head hidden layers | 1 | {0, 1, 2, 3} | Categorical |
| Adversarial Training | False | {True, False} | Categorical |
| **Season** | | | |
| Loss Weight | 0.9552523645401945 | (0, 1.5) | Linear |
| Prediction head activation | relu | {"relu", "swish", "gelu"} | Categorical |
| Prediction head hidden dimension | 512 | {64, 128, 265, 512} | Categorical |
| Prediction head hidden layers | 2 | {0, 1, 2, 3} | Categorical |
| Adversarial Training | False | {True, False} | Categorical |

## A.10. Multitask learning formalism

Let $\mathcal{D} = \{(x_i, y_i, \{m_i^k\}_{k=1}^K)\}_{i=1}^N$ denote the training dataset (post mixup), where $x_i$ is an input audio spectrogram (window), $y_i \in \mathcal{Y}$ is the primary target label corresponding to the annotated focal species, $m_i^k$ is the target for metadata source $k$ out of total $K$ metadata sources. If a particular metadata $k$ is absent for the datapoint $i$, we set its target to a placeholder value $m_i^k = \phi$.

We encode the audio spectrogram with a shared encoder $f_\theta(\cdot)$. The species prediction head $g_\phi$ then predicts the focal species as

$$\hat{y}_i = g_\psi(f_\theta(x_i)), \qquad (4)$$

where $\psi$ are the species decoder parameters. Similarly, each metadata $k \in \{1, \ldots, K\}$ is associated with a decoder

$$\hat{m}_i^k = h_{\psi_k}^k(f_\theta(x_i)), \qquad (5)$$

with task-specific parameters $\psi_k$.

The overall multi-task objective combines the species identification loss used in Perch 2.0 (van Merriënboer et al., 2025) along with metadata-specific loss formulations $\{\mathcal{L}_k^m\}_{k=1}^K$ as:

$$\mathcal{L}_{\text{total}} = \mathcal{L}_{\text{Perch}}(\hat{y}_i, y_i) + \sum_{k=1}^K \lambda_k \mathbb{1}[m_i^k \neq \phi]\mathcal{L}_k^m(\hat{m}_i^k, m_i^k),$$

$$(6)$$

where $\{\lambda_k\}_{k=1}^K$ control the relative importance of the different metadata, and are treated as hyperparameters to be optimized by Vizier. Metadata specific preprocessing such as conversion to S2Cell (Google, 2025) as well as other design choices such as the decision between standard or adversarial training is abstracted away in the loss functions $\{\mathcal{L}_k^m\}_{k=1}^K$.

*Table A4.* **BEANS benchmark results.** Species identification performance on BEANS (mean and standard deviation across 5 runs). Shorthands: Humbug → HumbugDB

| Method | ESC50 | Watkins | Bats | CBI | Dogs | Humbug | Speech Cmds | Mean |
|---|---|---|---|---|---|---|---|---|
| | | | | Classification (Accuracy) | | | | |
| BioBaseline | 0.856 ± 0.02 | 0.884 ± 0.017 | 0.804 ± 0.023 | 0.777 ± 0.005 | 0.944 ± 0.025 | 0.786 ± 0.005 | 0.773 ± 0.028 | 0.832 ± 0.012 |
| METAPERCH | 0.879 ± 0.011 | 0.905 ± 0.003 | 0.816 ± 0.003 | 0.771 ± 0.004 | 0.961 ± 0.009 | 0.798 ± 0.006 | 0.778 ± 0.007 | 0.844 ± 0.002 |

*Table A5.* **BEANS benchmark results.** Species identification performance on BEANS (mean and standard deviation across 5 runs). Shorthands: ENA → ENABirds, Gibbon → Hainan Gibbons.

| Method | DCASE | ENA | Hiceas | RFCX | Gibbons | Mean |
|---|---|---|---|---|---|---|
| | | | Detection (cMAP) | | | |
| BioBaseline | 0.477 ± 0.010 | 0.778 ± 0.012 | 0.563 ± 0.020 | 0.196 ± 0.012 | 0.519 ± 0.030 | 0.506 ± 0.011 |
| METAPERCH | 0.496 ± 0.005 | 0.787 ± 0.003 | 0.568 ± 0.023 | 0.209 ± 0.009 | 0.500 ± 0.020 | 0.512 ± 0.008 |

*Table A6.* **BirdSet benchmark results.** Species identification performance on BirdSet (mean and standard deviation across 5 runs).

| Method | PER | NES | UHH | HSN | NBP | SSW | SNE |
|---|---|---|---|---|---|---|---|
| | | | | ROC-AUC | | | |
| BioBaseline | 0.756±.009 | 0.942±.004 | 0.874±.013 | 0.893±.017 | 0.935±.002 | 0.971±.002 | 0.868±.008 |
| METAPERCH | 0.801±.007 | 0.948±.003 | 0.900±.008 | 0.909±.017 | 0.937±.005 | 0.971±.001 | 0.874±.005 |
| | | | | cmAP | | | |
| BioBaseline | 0.217±.008 | 0.412±.003 | 0.344±.003 | 0.538±.012 | 0.692±.007 | 0.472±.014 | 0.349±.002 |
| METAPERCH | 0.261±.007 | 0.396±.003 | 0.356±.006 | 0.531±.017 | 0.698±.009 | 0.473±.004 | 0.353±.003 |
| | | | | Top 1 Accuracy | | | |
| BioBaseline | 0.500±.013 | 0.558±.014 | 0.651±.013 | 0.625±.022 | 0.717±.009 | 0.784±.005 | 0.792±.012 |
| METAPERCH | 0.559±.016 | 0.545±.005 | 0.550±.016 | 0.58±.014 | 0.679±.025 | 0.795±.004 | 0.778±.005 |

*Table A7.* **WABAD results.** Species identification performance on WABAD (mean and standard deviation across 5 runs).

| Method | 1-shot ROC-AUC | supervised | Proto ROC-AUC |
|---|---|---|---|
| BioBaseline | 0.912 ± 0.003 | 0.741 ± 0.027 | 0.928 ± 0.002 |
| METAPERCH | 0.917 ± 0.003 | 0.811 ± 0.015 | 0.946 ± 0.001 |

*Table A8.* **WABAD results.** Species identification performance on WABAD (mean and standard deviation across 5 runs).

| Method | Boreal Forests/Taiga | Deserts / Xeric Shrublands | Wetlands | Mediterranean |
|---|---|---|---|---|
| BioBaseline | 0.892 ± 0.004 | 0.936 ± 0.004 | 0.887 ± 0.006 | 0.917 ± 0.006 |
| METAPERCH | 0.910 ± 0.004 | 0.912 ± 0.007 | 0.899 ± 0.003 | 0.929 ± 0.002 |

| Method | Temperate | (sub) Tropical | Montane And Savannas | Unspecified |
|---|---|---|---|---|
| BioBaseline | 0.921 ± 0.007 | 0.916 ± 0.001 | 0.879 ± 0.011 | 0.793 ± 0.007 |
| METAPERCH | 0.926 ± 0.003 | 0.939 ± 0.001 | 0.906 ± 0.007 | 0.810 ± 0.004 |

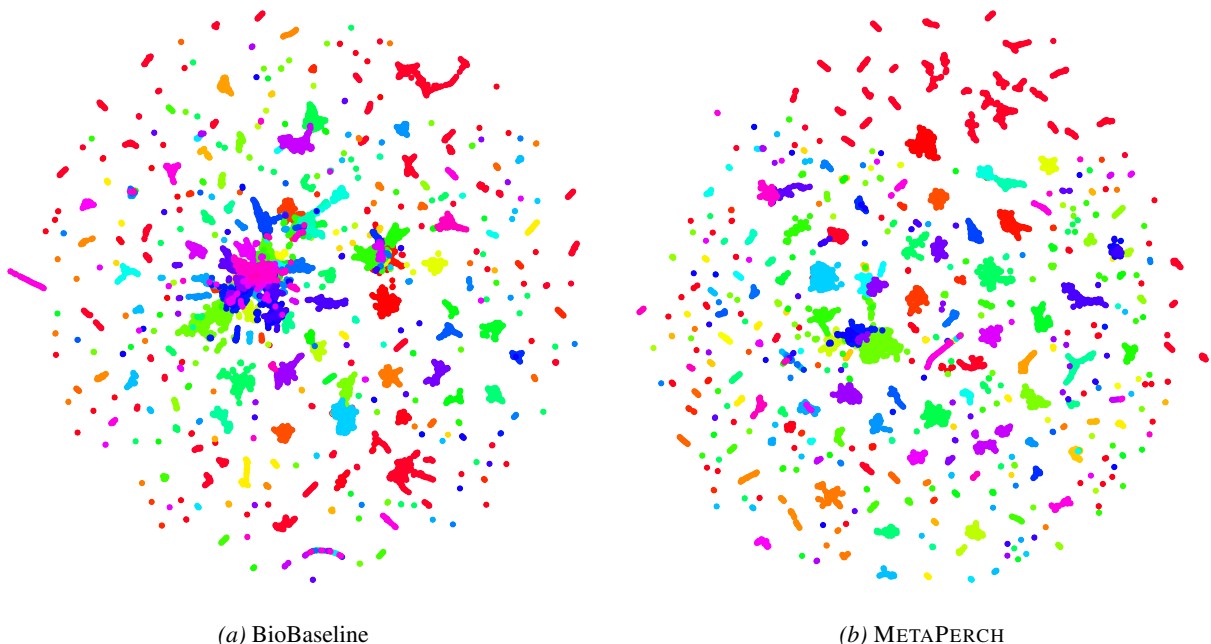

*(a)* BioBaseline            *(b)* METAPERCH

*Figure A6.* **WABAD UMAP plots—location.** Embeddings of windows are colored by recording site (there are 72 recording sites in total).

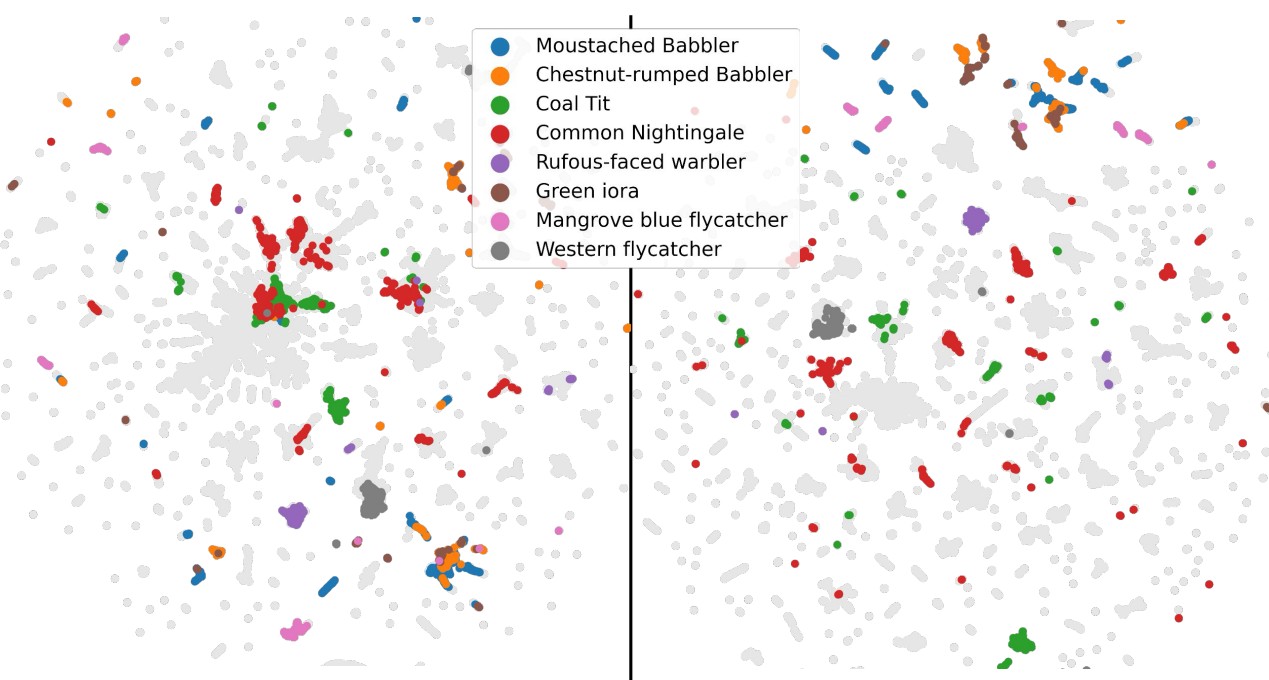

*Figure A7.* **WABAD UMAP plots—select species.** Embeddings of windows are colored by the species, for a few specific species Left: BioBaseline. Right: METAPERCH.

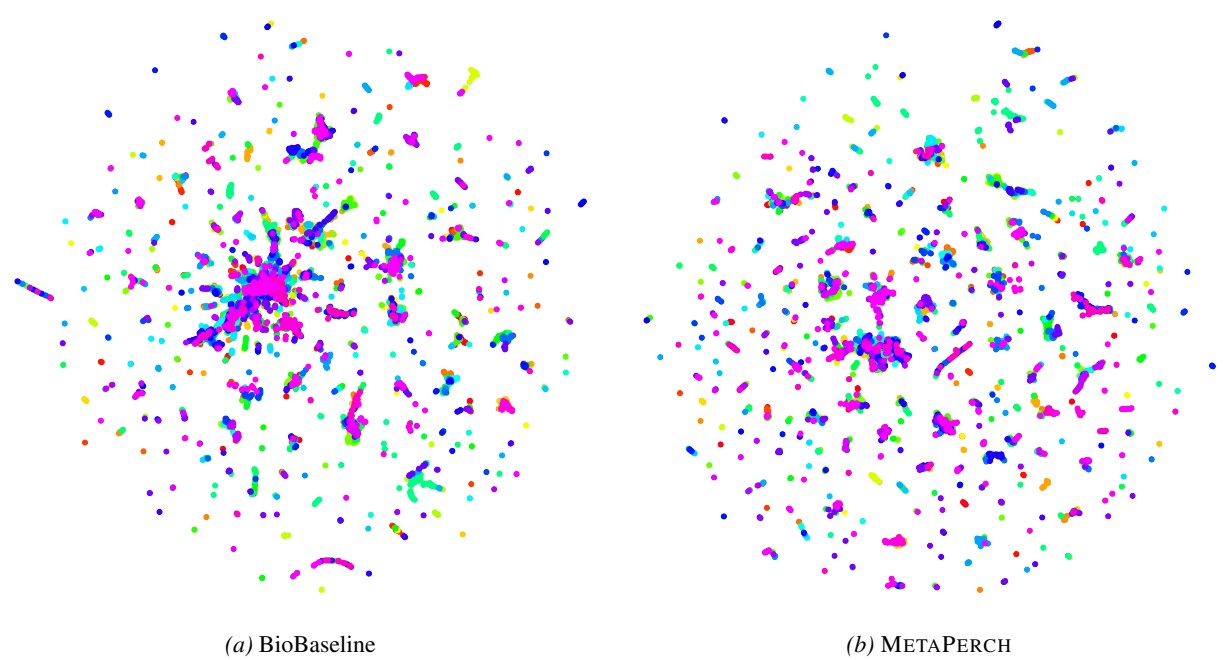

*(a)* BioBaseline

*(b)* METAPERCH

*Figure A8.* **WABAD UMAP plots—all species.** Embeddings of windows are colored by the species in alphabetical order of their scientific names (so sibling species may have similar colors). The windows are filtered to those that only have a single species.

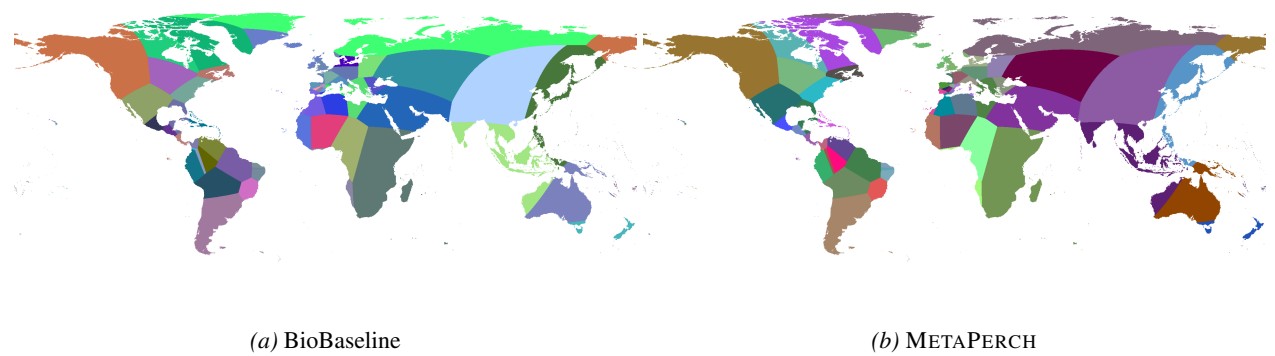

*(a)* BioBaseline

*(b)* METAPERCH

*Figure A9.* **WABAD UMAP plots.** Color mappings between BioBaseline and METAPERCH are independent of one another (but are normalized using the same normalization constants). There is semantic meaning with respect to the colors in each plot: similar colors reflect embedding similarity.

