# OpenReview forum: "MetaPerch: Learning from metadata for bioacoustics foundation models"
_ICML.cc/2026/Conference — ICML 2026 regular_

### Official Review · Reviewer_TsJV · 2026-03-10

**Soundness:** 3
**Presentation:** 3
**Significance:** 3
**Originality:** 3
**Overall Recommendation:** 4
**Confidence:** 2

**Summary:**

The paper introduces METABIO, a bioacoustic foundation model that uses recording metadata (like location) as auxiliary supervision targets during training. The authors frame species detection alongside metadata prediction as a multi-task learning problem. The architecture follows previous work with additional heads for metadata prediction. The authors demonstrate that incorporating these auxiliary signals improves the performance across a set of benchmarks.

**Compliance With Llm Reviewing Policy:**

Affirmed.

**Key Questions For Authors:**

1. Line 221: The approach used leads to mislabelling metadata, but it still improves the performance. Can the authors comment why they think this might happen?
2. From Table 6c, it seems like seasonal metadata does not matter much as the adversarial approach is almost similar, but it is clear that location and background matter. So, it is weird that the final setting uses season as well. Do the authors have some comments regarding this conflict?
3. As background species metadata is very limited, how much does its presence affect the performance? Do the authors have an ablation by removing this metadata?

**Limitations:**

See weakness please

**Strengths And Weaknesses:**

# Strengths
1. The authors successfully extract value from metadata that is naturally available but historically ignored in standard species-only learning paradigms.
2. The authors use a practical solution for missing metadata by assigning placeholder values and zeroing out their specific loss contributions, allowing the model to benefit even when metadata availability drops.
3. While heavily trained on bird data, METABIO shows good transfer capabilities to non-avian recognition tasks. It achieved accuracy improvements on the Watkins marine dataset and the Bats dataset.

# Weakness
1. The use of a small, vision backbone (EfficientNet-B3) to process spectrograms may limit the model's capacity. Using a more modern machine perception architecture, or using encoders from Multi-modal large language models (MLLMs) that natively understand audio-text relationships, could potentially yield richer semantic alignments than simple MLP projection heads.
2. The performance improvements from geographic metadata are inherently tied to the biases in citizen science platforms (e.g., the massive overrepresentation of North America and Western Europe).
3. The recorded gains are quite minimal in the BirdSet compared to Perch2.0.

---

> ### Author Rebuttal · Authors · 2026-03-30
>
> Thank you for your thoughtful and constructive feedback. We appreciate your recognition of the research gap in past bioacoustics research that our work addresses. We thank you for highlighting the transfer capabilities of our models to non-avian tasks and are glad that you found our approach to handling missing metadata practical.
>
> Below, we address specific concerns:
>
> **[W1] Small vision backbone**
>
> The current SotA, Perch 2.0, also uses EfficientNet-B3 as its backbone. Because our objective is to investigate the influence of metadata on bioacoustics tasks, we use the same backbone as Perch 2.0, and compare model performance with and without metadata auxiliary losses. This choice is also supported by past bioacoustics research demonstrating that smaller models tend to generalize better to domain shifts in bioacoustics evaluation setups then bigger transformer-based backbones (Hamer et al., 2023).
>
> While MLLMs offer richer semantic audio-text alignments, they are often heavily biased towards speech. In our experience, their applicability on our downstream bioacoustics tasks is limited, as they often do not encode fine grained details required for distinguishing species sounds. MLLMs trained specifically for bioacoustics may be promising alternatives, but they have yet to achieve SotA on current bioacoustics benchmarks—for example, Perch 2.0's BEANS performance is considerably better than NatureLM-audio's. Specifically for our investigations, we note that MLLMs often already incorporate some metadata in the captions used for training, making it more challenging to study the disentangled influence of different metadata in isolation.
>
> **[W2] Location metadata improvements tied with geographic biases in training data**
>
> Yes, the improvements from metadata are tied to geographic biases as we observe greater performance improvements for underrepresented regions. However, we see this as a strength rather than weakness. Mitigating the effects of these geographical biases in citizen science datasets was one of the primary motivations for including metadata auxiliary losses. In the absence of geographical biases, metadata is still useful in impactful applications that require generalization to species and acoustic domain shifts.
>
> **[W3] Marginal gains over Perch 2.0 for BirdSet**
>
> While we acknowledge that the gains of MetaBio over Perch 2.0 on BirdSet are minimal, we wish to stress that the gains in comparison with BioBaseline are more significant. This comparison highlights the benefit of metadata, which is the primary focus of our work.
>
> **[Q1] Metadata mislabeling**
>
> In the case of mixing recordings with missing metadata, the alternative to mislabeling is the exclusion of available metadata. This observation suggests that our method demonstrates some robustness to noise in the metadata labels and that noisy metadata labels are more beneficial than no labels.
>
> There are also a few cases where the mixed metadata would actually be labeled close to correct:
>
> 1. The true value of missing metadata is similar to other metadata being mixed
>    (e.g. **mixing recordings from similar regions** does not lead to significant mislabeling of location, even if the metadata is missing for some).
> 2. A missing categorical metadata is actually absent
>    (e.g. many recordings without the background species metadata **may not actually have any background species**, so their mixing does not lead to mislabeling).
> 3. The signals in a recording with a missing metadata are agnostic to the metadata
>    (e.g. **musical instrument sounds** from FSD50K are generally **independent of location and time**, so assigning the mixed location or time to these doesn’t yield a semantic mislabeling).
>
> **[Q2] Reasoning for inclusion of season metadata**
>
> This is a fair question. We performed model selection using validation performance, which can exhibit gaps or generalization shifts with the evaluation setup (thus showing different trends than eval performance). Please refer to our response to WRFF **[Q2]** for more on this disparity.
>
> Beyond performance gains, our choice also reflects strong domain priors about the utility of season as a metadata. Many species activities—such as migration and mating—are correlated with seasonality, and including this context could help the models learn some useful correlations. For migratory species in particular, the season could provide useful context complementary to the geographic location.
>
> **[Q3] Effect of background species metadata**
>
> The effect of background species is apparent in our individual metadata experiments in Figure 6c. Despite limited availability, it yields one of the highest performance gains among the metadata we experiment with. The benefit of sparse, but highly informative metadata, is further corroborated by our experiments that model the effect of missing metadata rates in Figure 6b—even small amounts of metadata presence can yield significant performance boosts.

---

> > ### Author Rebuttal · Reviewer_TsJV · 2026-04-03
> >
> > I appreciate the authors response. For the mislabeling of the metadata, the argument that 'noisy labels are better than no labels' is an interesting observation, the response remains largely qualitative. Specifically, the claim that the model is robust to these incorrect assignments lacks a controlled ablation study, so I am not fully convinced that this approach is universally beneficial across more complex domains. Therefore, I will keep my score unchanged.

---

> > > ### Author Response · Authors · 2026-04-07
> > >
> > > Thank you for considering our rebuttal.
> > >
> > > Please allow us to clarify our points regarding metadata mislabeling during mixup. We analyse the effect of the handling of missing metadata during mixup in our ablations Figure 6e (rows repeated below for easier reference)
> > >
> > > | Method | WABAD | Val set |
> > > | ---- | ---- | ---- |
> > > | MetaBio | 0.946 | 0.880 |
> > > | Mix iff present | 0.945 | 0.874 |
> > >
> > > The first row presents the performance of MetaBio, where we mix any available metadata, which could potentially lead to mislabeling. The second row is the alternative, where the corresponding metadata is marked absent if any mixed metadata is absent (we call it ‘mix if and only if all present’). Our comment in the rebuttal was in reference to this experiment: we observe that using potentially mislabeled metadata yields better performance than treating them as absent/missing (not using them). The performance difference is relatively small, however, suggesting that the model exhibits some robustness to this design choice.
> > >
> > > In the revised version, we will elaborate on this discussion in the ablations section and include a forward reference to this discussion in section 3, where we actually describe the design choice.

---

### Official Review · Reviewer_ALaD · 2026-03-11

**Soundness:** 3
**Presentation:** 2
**Significance:** 2
**Originality:** 3
**Overall Recommendation:** 2
**Confidence:** 3

**Summary:**

This paper proposed a new bioacoustics foundational model for species identification called METABIO to highlight the influences of metadata as auxiliary targets.

The METABIO model was constructed using a two-step process: first, establishing a baseline model (which is referred as "BioBaseline" in the paper) by modifying the existing bioacoustics foundational models for faster iteration and better exposure to metadata's influence; second, incorporating metadata prediction tasks as MLPs into the baseline models and select the best performing model.

The paper also conducted comparisons between METABIO and BioBaseline against several benchmarks.

The authors claimed that the metadata introduction into bioacoustics foundational models improved species identification.

**Compliance With Llm Reviewing Policy:**

Affirmed.

**Final Justification:**

The paper studies a relevant and practically meaningful problem: whether recording metadata can be used as auxiliary supervision to improve bioacoustic species identification. I acknowledge several strengths of the submission. The paper is clearly written, the empirical evaluation is broad, and the experimental study is careful and substantial. In particular, the authors evaluate across many benchmarks and provide useful ablations on different metadata sources and design choices. These aspects make the work sound at an empirical level and potentially useful to the community.

At the same time, my final recommendation is negative because I remain unconvinced by the paper’s originality and overall significance for ICML. In my view, the core method is a relatively straightforward multitask extension of an existing baseline: metadata are incorporated through auxiliary prediction heads, losses are balanced empirically, and the final design is selected through model selection over metadata/task choices. While this is a reasonable empirical approach, I do not see a sufficiently strong methodological advance beyond combining known ingredients in this application setting.

My main concern in the original review was the lack of stronger theoretical or principled justification for the choice of metadata and for their claimed influence on transfer/generalization. After reading the rebuttal, I acknowledge that the authors clarified their qualitative intuition and pointed to the relevant discussion in the paper. However, this response did not resolve my main concern. The rebuttal provides plausible motivation, but not a more systematic account of when auxiliary metadata prediction should help, when it might hurt, or why the selected metadata are preferable beyond empirical performance. As a result, the rebuttal did not increase my confidence in the core contribution, and in fact reinforced my view that the paper’s main value is empirical rather than methodological.

In weighing the dimensions: I view the paper as reasonably sound empirically and fairly clear in presentation, with some minor presentation issues that are likely fixable. However, I assess originality and significance less positively, because the contribution appears limited mainly to empirical integration and benchmarking rather than a stronger conceptual or algorithmic advance. For that reason, and after considering the rebuttal, I lower my recommendation from 3 to 2. The work has merits, but I remain below the acceptance bar for ICML.

**Key Questions For Authors:**

1. Do you have any theory to support your selection of metadata?
2. Do you have any theory or proof to show the influence of metadata on the outcome systematically?

**Limitations:**

Yes

**Strengths And Weaknesses:**

Strengths:

 1. The introduction of metadata as auxiliary targets is creative
 2. The paper conducted a strong empirical study
 3. The influence of metadata is clearly explained in each benchmark used
 4. Well written

Weakness:

 1. The presentation of tables and graphs were inconsistent and mixed up, with some violations of ICML paper format (i.e. included small titles in graph, combining tables and graphs into one figure in Figure 2, 6 and Table 3)
 2. The selection of metadata lacks theoretical/ qualitative reasoning
 3. The influence of metadata on the results lacks theoretical reasoning

---

> ### Author Rebuttal · Authors · 2026-03-30
>
> Thank you for your thoughtful and constructive feedback. We appreciate your recognition of the creativity of our approach, empirical strength of our study, and writing style. We are glad that you found our metadata influence discussions clear and useful.
>
> Below, we address specific concerns:
>
> **[W1] Presentation of tables and graphs**
>
> To our knowledge, we are not violating any explicit paper formatting requirements. For the camera ready, we're very happy to make adjustments to improve the accessibility of our tables and figures. We will increase the font sizes of our graph titles in Figures 4, 6a, and 6b, and Table 3-right. We will make the distinction between tables and graphs more explicit in Figure 2 and Table 3. For Figure 6, we will put panels (a), (b), (c), and (f) into one figure and panels (d) and (e) into a separate table for more clarity.
>
> **[W2-3 + Q1-2] Theoretical/qualitative reasoning for metadata selection and influence**
>
> We include qualitative reasoning for metadata influence and selection in Section 3.3, lines 190-201, focusing on how each of location, season, and background species would intuitively help address specific challenges in our evaluation tasks, such as species, acoustic, and geographic domain shifts. We believe that these arguments, together with our empirical results, provide sufficient justification for our key claims about metadata selection and influence.

---

> > ### Author Rebuttal · Reviewer_ALaD · 2026-04-03
> >
> > Thank you for the rebuttal and for the clarifications. I appreciate the authors’ response regarding the presentation issues; these seem minor and likely fixable in a revision.
> >
> > However, my main concerns remain unresolved. In response to my questions about the selection of metadata and the influence of metadata on the outcome, the rebuttal mainly points to qualitative intuition already present in the paper (e.g., location, season, and background species being plausibly correlated with species identification). I agree that this provides reasonable motivation, but it does not substantially strengthen the paper’s theoretical or methodological contribution.
> >
> > My core concern is that the paper’s main technical contribution remains a relatively straightforward multitask extension of an existing baseline, where metadata are added as auxiliary prediction heads and the final design is chosen largely through empirical selection. The rebuttal does not provide a more principled account of why these metadata should systematically improve transfer/generalization, nor does it clarify when such auxiliary signals may help versus reinforce spurious correlations. Because this concern goes to the central contribution of the work, I do not think it can be fully addressed within a short rebuttal.
> >
> > As a result, after considering the rebuttal, I lower my assessment from weak reject to reject. While I still see value in the empirical study and acknowledge the paper is well written, I remain unconvinced that the methodological novelty and justification are sufficient for acceptance at ICML.

---

> > > ### Author Response · Authors · 2026-04-07
> > >
> > > Thank you for considering our rebuttal.
> > >
> > > As the Main Track Reviewer Form Instructions indicate regarding novelty, "originality does not necessarily require introducing an entirely new method. Rather, a work that provides novel insights by evaluating existing methods, or demonstrates improved understanding is also equally valuable."
> > >
> > > We firmly believe in the value of our submission in that respect, especially in a field (bioacoustics) that is anchored in concrete downstream applications and assigns a high value to quantitative evidence. As a reminder, to the best of our knowledge we are the first to evaluate the individual contributions of a large number of metadata sources (9) on a comprehensive bioacoustics evaluation suite covering 17 downstream datasets. Through our experiments, we find the metadata that are most useful and highlight the scenarios where these metadata do and do not improve performance.
> > >
> > > The methodological novelty of our work comes from the application of leveraging untapped metadata in building a foundational bioacoustics model. A benefit of our proposed approach is in its simplicity: it is robust to missing metadata and allows for end-user flexibility while providing high predictive performance comparable to or exceeding much more complex modeling approaches (NatureLM-Audio).
> > >
> > > Beyond that, we're demonstrating the utility of metadata in building a robust, generalizable foundational bioacoustics model. This presents a lower-bound on the potential for this training paradigm within this field and provides a strong case to advocate for the inclusion of metadata in crowdsourced, citizen science platforms and large-scale data collection.
> > >
> > > As for the expectation of a principled or theoretical justification—in addition to the proposed method and empirical contributions—for why the selected metadata should help, we believe this does not detract from our work's strengths across the four dimensions (soundness, presentation, significance, and originality) in the reviewer instructions.

---

### Official Review · Reviewer_WRFF · 2026-03-12

**Soundness:** 3
**Presentation:** 3
**Significance:** 2
**Originality:** 2
**Overall Recommendation:** 2
**Confidence:** 3

**Summary:**

The paper proposes MetaBIO a bioacoustic foundation model for species identification by leveraging recording metadata as auxiliary supervision. The main contribution of the paper is exploring the addition of metadata such as geographic location, season and background species for learning more robust features. Extensive testing on 17 bioacoustic datasets reveals that MetaBIO achieves state-of-the-art performance, particularly in underrepresented regions and for novel species. The paper also explores various design choices, including adversarial training to mitigate spurious correlations and methods for handling missing metadata. The work demonstrates that global taxonomic patterns found in metadata can significantly bridge the gap in biodiversity monitoring applications.

**Compliance With Llm Reviewing Policy:**

Affirmed.

**Final Justification:**

The paper has relatively low technical novelty and is a straightforward extension of Perch 2.0 to a multitask setup. The paper claims to be the first to incorporate audio metadata into the training, however misses several works which have already incorporated metadata into their framework. The paper needs to expand the related works section and discuss similarities and differences with works that incorporate metadata such as [1, 2]. On top of it, I have some concerns regarding the transfer of model to newer tasks. The model overfits on the validation tasks and fails to transfer well to new tasks. The authors did not address this problem in the paper and leave it as a future work. Given these concerns, the paper would benefit from more careful design and experiments. At present, I would like to lower my score from 3 to 2.

References

[1] Khanal, Subash, et al. "Psm: Learning probabilistic embeddings for multi-scale zero-shot soundscape mapping." Proceedings of the 32nd ACM International Conference on Multimedia. 2024.

[2] Khanal, Subash, et al. "Sat2Sound: A Unified Framework for Zero-Shot Soundscape Mapping." arXiv preprint arXiv:2505.13777 (2025).

**Key Questions For Authors:**

1. Can you include experiments that confirm the gains reported in the paper are due to the proposed methodology and not the chosen training dataset? For example, can you provide results for fine-tuned versions of Perch 2.0?
2. In Figure 6d, why does the median Vizier trial outperform the best-validation trial on WABAD?

**Limitations:**

Limitations are discussed in the paper.

**Strengths And Weaknesses:**

**Strengths**

1. The paper covers 17 evaluation datasets across multiple taxa and task types. This makes the paper more than a toy demonstration.
2. The main strength of the paper is in depth of experiments rather than the technical methodological contributions, which are important from an empirical viewpoint.
3. The paper is well written, structured and offers honest insights into the results.

**Weaknesses**

1. Although experiments are extensive, the methodology and results seem quite incremental. Compared to Perch 2.0, it seems like an incremental paper without concrete technical contributions. Several works in the past have already proposed incorporating metadata into classification problems in various ways.
2. I am not able to clearly differentiate the advantages of incorporating metadata compared to training on a large dataset, since the BioBaseline also outperforms Perch 2.0, in some cases. What happens if you fine-tune perch 2.0 on the same dataset used in the paper with/without adding metadata. This experiment will help disentangle the gains coming from dataset scale and metadata inclusion.
3. Since the paper is proposing a foundation model, the authors should also compare various backbone architectures and backbone sizes. Experimenting on a single backbone is not enough to claim the model being a foundation model.
4. The paper claims metadata provides robust representation for the model to learn. However, BioBaseline performs better than MetaBio in high shot settings as reported in Figure 6a. The paper should include more discussion on this front.
5. The paper should provide an uncertainty analysis of the reported results against metadata. Metadata can be highly unreliable for a lot of reasons. Errors in location can vary from meters to kilometers. How does the model handle such kind of uncertainties in the metadata?

---

> ### Author Rebuttal · Authors · 2026-03-30
>
> Thank you for your thoughtful and constructive feedback. We are glad that you found our paper to be well written, structured and offering honest insights. We appreciate your recognition of the depth of our experimental setup and its broad coverage of multiple taxa and task types.
>
> Below, we address specific concerns:
>
> **[W1] Difference from past works incorporating metadata into classification problems**
>
> We discuss how we differ from past bioacoustics research with metadata in our related work. The key differences are summarized here:
> 1. Unlike methods that condition on metadata (Jeantet & Dufourq 2023), we use metadata to form auxiliary training losses, thereby relaxing the assumption that metadata is available at test time. This facilitates broader downstream use and applications.
> 2. Instead of using species-level metadata such as sound descriptions and functional traits (Gebhard et al 2024; Tobias et al 2022), we use recording-level metadata that provide additional context complementary to species information.
> 3. In place of encoding various recording metadata into a single feature (like NatureLM-audio and AnimalSpeak), we use each metadata to construct individual prediction tasks which allow us to disentangle their contributions and effects on species detection.
>
> To the best of our knowledge, we are the first to systematically investigate the individual contributions of a large number of metadata sources (9) on a comprehensive bioacoustics evaluation suite. We believe that using standardized metadata co-training methods from other domains is a good first step that paves the way for future research into more involved metadata-aware approaches tailored for bioacoustics.
>
> **[W2 + Q1] Incorporating metadata vs dataset scale**
>
> We use the same training dataset as Perch 2.0 (refer to Section 3.1) to train both BioBaseline and MetaBio, so there is no difference in dataset scale. The only difference between BioBaseline and Perch 2.0 is the exclusion of source prediction and self-distillation. The performance gains we observe between MetaBio and BioBaseline reflect the inclusion of metadata prediction tasks only (this is the only difference between the two approaches).
>
> **[W3] Foundation model claim and alternate backbones**
>
> Foundation models are not characterized by their applicability to various backbone architectures, but rather strong performance and robustness across a broad range of downstream applications. We demonstrate strong performance and robustness of our model through large-scale experiments on 17 evaluation datasets across a wide range of taxa and task types. We believe these empirical analyses are sufficient to demonstrate the broad downstream utility, and therefore the foundational nature, of MetaBio.
>
> **[W4] BioBaseline outperforms MetaBio in BIRB high shot settings**
>
> Our BIRB experiments include 2 settings: Xeno-Canto held-out and PAM soundscapes. With high shots, MetaBio performs worse than the baseline for the former, and the performance is about the same for the latter. The Xeno-Canto held-out set reflects an academic/laboratory-esque setting designed to artificially disentangle the effects of acoustic and species shifts. PAM soundscape experiments are more representative of natural deployment conditions. While we include both settings to demonstrate holistic behavior instead of cherry-picking our results, we expect PAM soundscape results to be more reflective of actual metadata benefit in practice. We will emphasize this in our camera ready version.
>
> **[W5] Metadata uncertainties**
>
> We evaluate performance on some dimensions of metadata uncertainty, namely different rates of missing metadata (refer to Figure 6b). These results show that while the performance decreases as less metadata is present, there are still notable improvements over the baseline for as low as 1% availability.
>
>
> MetaBio implicitly handles small uncertainties and noises in metadata because of the way each prediction task is formulated. For location, we aggregate all locations within neighborhoods of roughly 5,000 sq km into a single S2 cell (level 7), and the model ought to be robust to uncertainties in location up to this threshold. Similarly, for the date and time-of-day, we bucket the metadata into broad categories (4 seasons and 4 day-parts), and any variation within each category should not have any effect on model training.
>
> **[Q2] Median Vizier trial outperforms best Vizier trial on WABAD**
>
> Since we evaluate on new tasks altogether, our model selection criterion necessarily differs from our evaluation criterion. This performance disconnect is a classic case of overfitting on the validation tasks, and we cannot retrospectively use the median Vizier trial based on evaluation performance. The inclusion of this result in our ablations was intended to expose this very observation and highlight the challenges of choosing appropriate validation tasks; we will clarify this in the camera-ready.

---

> > ### Author Rebuttal · Reviewer_WRFF · 2026-04-03
> >
> > Thank you very much for the rebuttal. The rebuttal addresses some of my concerns. Below are remaining concerns:
> >
> > 1. **Novelty**: There are works which use metadata to form auxiliary losses [1]. Also, there are two recent works which incorporate audio source into their framework [2, 3]. These works should be discussed.
> >
> > 2. **Overfitting**: Why do the authors call this a foundation model when it does not transfer well to the evaluation datasets. Seems like the training/validation dataset is not representative enough to transfer to real world benchmarks.
> >
> > **References**
> >
> > [1] Ayush, Kumar, et al. "Geography-aware self-supervised learning." Proceedings of the IEEE/CVF international conference on computer vision. 2021.
> >
> > [2] Khanal, Subash, et al. "Psm: Learning probabilistic embeddings for multi-scale zero-shot soundscape mapping." Proceedings of the 32nd ACM International Conference on Multimedia. 2024.
> >
> > [3] Khanal, Subash, et al. "Sat2Sound: A Unified Framework for Zero-Shot Soundscape Mapping." arXiv preprint arXiv:2505.13777 (2025).

---

> > > ### Author Response · Authors · 2026-04-07
> > >
> > > Thank you for considering our rebuttal; we are glad that we were able to address some of your concerns.
> > >
> > > 1. **Novelty**
> > >
> > > Thank you for bringing these papers to our attention. While [1] uses metadata to form auxiliary losses, this work is within the context of remote sensing (satellite imagery), a substantially different problem setting from bioacoustics—a field where the use of metadata is under-explored. Furthermore, while their metadata is restricted to location (and some notion of time), our work includes a systematic study of 9 metadata sources and their effects on 17 bioacoustics evaluation datasets across multiple taxa and task types.
> > >
> > > We also appreciate you pointing out the 2 soundscape mapping papers [2, 3]. While our current discussion of related work primarily focuses on past bioacoustics research, we will be glad to include an additional section discussing similar approaches in other domains such as general audio and images in our camera ready.
> > >
> > > 2. **Overfitting**
> > >
> > > Please allow us to clarify our comment about overfitting on the validation tasks. As you point out, validation tasks that are not representative enough of evaluation tasks could lead to potential overfitting, which we observe in the disconnect between our validation and WABAD performances. Since we use the same standard training/validation setup as past bioacoustics foundation models (Perch 2.0), this is a systemic challenge in bioacoustics, rather than a weakness of our method. By reporting both validation and evaluation performances, we simply wished to highlight this challenge that may have been overlooked by past works that only perform ablations on the validation tasks.
> > >
> > > Regarding the transfer to real-world evaluation benchmarks, we wish to stress that MetaBio yields results on-par or better than Perch 2.0, a strong bioacoustics foundation model. MetaBio achieves state-of-the-art performance on BEANS classification and detection tasks, out-performing Perch 2.0 by 2.3% and 1.0% respectively. On BirdSet, it achieves similar performance as Perch 2.0 (0.2% lower ROC-AUC and 0.7% higher cmAP). Strong performance on these benchmarks—covering a broad range of taxa and tasks—supports the ability of MetaBio to transfer to real-world deployment settings.
> > >
> > > ---
> > >
> > > We hope that our response addresses your remaining concerns, and respectfully ask you to reconsider your score.

---

### Official Review · Reviewer_NBua · 2026-03-13

**Soundness:** 4
**Presentation:** 4
**Significance:** 3
**Originality:** 3
**Overall Recommendation:** 4
**Confidence:** 5

**Summary:**

The paper explores learning bioacoustics foundation models by leveraging available metadata such as location and time to exploit species-metadata correlations in their learned representations. The authors introduce MetaBio, a foundation model trained on a diverse collection of animal vocalizations and environmental sounds, considering 9 types of metadata. They evaluate MetaBio on 17 bioacoustic datasets, demonstrating that including metadata improves species identification on 16 out of 17 datasets.

**Compliance With Llm Reviewing Policy:**

Affirmed.

**Final Justification:**

The study and the shared model are undoubtedly useful for the field of bioacoustics and for future machine learning research in this area. However, the contribution in terms of methodological foundations remains too modest for a conference such as ICML, which is why I am not raising my score.

**Key Questions For Authors:**

The last paragraph of Section 4 explaining Figure 6(f) is unclear. Could the authors provide a clearer explanation of this part?
Line 319, first column: it should be accuracy and not ROC-AUC.
Line 319, second column: values are reported as “1% accuracy” and “0.006 cmAP.” For consistency, please use a uniform format throughout the paper.

**Limitations:**

yes

**Strengths And Weaknesses:**

Strengths:

Soundness: Claims are well supported by experiments across 17 datasets.
Presentation: Clear, well-structured, easy to follow.
Originality:  The integration of metadata for species identification is not a new idea itself, but its integration into a foundation model is a relevant contribution.
Significance: MetaBio can help increase robustness of species identification tools to various domain shifts.

Weaknesses:

MTL paradigm is not formalized.
The choice of metadata prediction task is not explored; location metadata, for instance, could be used with a geoloc encoder from TaxaBind as a target, or with a strong Earth embedding such as AlphaEarth.

---

> ### Author Rebuttal · Authors · 2026-03-30
>
> Thank you for your thoughtful and constructive feedback. We are glad that you found our paper clear, well-structured, and easy to follow. We are pleased that you found our claims to be well supported by our experiments, and appreciate your positive assessment of our originality, contributions, and significance.
>
> Below, we address specific concerns:
>
> **[W1] MTL paradigm is not formalized**
>
> Multi-task learning has grown increasingly common in recent years. As such, we opted to focus only on textual and visual description of individual tasks to build the reader’s intuition. We will include additional details in the appendix to formalize the paradigm in our revised, camera ready version, with a focus on individual task formulations, loss weighting, and joint optimization.
>
> **[W2 + Q1] Choice of metadata prediction task and explanation of Figure 6f**
>
> We explore 8 different choices for the location prediction task in our ablations (Figure 6f), including regression (Cartesian and haversine), classification (different cell resolutions), and distillation with geolocation encoders such as GeoCLIP and SatCLIP. Beyond these, we also experimented with alternative formulations of season and time-of-day.
> Appendix `A.4: Additional method details` includes more details for each prediction task formulation we explored.
>
> We appreciate your suggestion of AlphaEarth as an alternative formulation for location prediction. Following a similar procedure as our GeoCLIP and SatCLIP experiments, we use AlphaEarth embeddings corresponding to the location and time of our recordings as distillation targets (trained with cosine loss). Results on WABAD are reported below.
>
> | Model | WABAD performance |
> | ---- | ---- |
> | BioBaseline |  0.928 |
> | BioBaseline + classification (level 7) | 0.946 |
> | BioBaseline + AlphaEarth | 0.942 |
>
> While AlphaEarth improves over the baseline, it falls short of classification with level 7, the formulation used in MetaBio. With these results, we plan to retain our location prediction formulation in MetaBio, but will include AlphaEarth results as part of our ablations in the camera ready version.
>
> Regarding the discussion of Figure 6(f) (lines 407-416, second column), we thank you for bringing it to our attention. In our camera ready version, we will improve its clarity and point to additional details in the appendix.
>
> **[Q2] Incorrect metric and result reporting format**
>
> We apologize for this mistake and the inconsistencies in the format of reported values, and appreciate you bringing them to our attention. In the camera-ready, we will correct the metric to accuracy and use fractions (0.01 instead of 1%) to refer to performances throughout the text.
>
> **References**
>
> [Weyand et. al] Weyand, Tobias, Ilya Kostrikov, and James Philbin. "Planet-photo geolocation with convolutional neural networks." European conference on computer vision. Cham: Springer International Publishing, 2016.

---

> > ### Author Rebuttal · Reviewer_NBua · 2026-04-03
> >
> > The authors carefully addressed my recommendations and questions.

---

### Decision · Program_Chairs · 2026-04-30

**Decision:**

Accept (regular)

**Comment:**

This paper proposes a training strategy for bioacoustic foundation models that enables metadata to be used during training as additional signal, and reflects the reality that not all bioacoustic data has metadata available. The benefits of this strategic and realistic integration of metadata are explored for a diverse set of downstream bioacoustics tasks from the BirdSet benchmark, and the the paper demonstrates that they have generated systematic training procedures that work well under realistic constraints regarding metadata availability, and balancing different availabilities and structures of metadata. The reviews for the work are mixed, with two weak accepts and two rejects. The reviewers emphasized that the proposed method extracts value from metadata to improve representation learning in bioacoustics, despite heterogeneous metadata types and availability, as well as appreciated the sound evidence of claims demonstrated experimentally on 17 downstream bioacoustics tasks, as well as the well-written and well-structured narrative. Some reviewers did express concerns that the methodological novelty was limited, requested additional comparisons across architectures, particularly more modern/larger architectures, and requested additional investigation into the demonstrated lower performance when fine-tuning on more data (high-shot settings), and potential lack of generalizability beyond validation sets. The generalizeability was addressed in the rebuttal, by emphasizing transfer to significantly OOD tasks is a systemic challenge faced by "foundation models" in bioacoustics based on the available training data, but they are still useful and highly-used. One reviewer thought there was lack of sufficient reference to prior work in acoustics that uses metadata, The ACs agree with the authors that the prior work referenced by the reviewer is not the same methodologically, but also emphasize that it should be cited/discussed to better illustrate what sets this work apart. After careful consideration, the ACs recommend acceptance of this work, as it represents a useful and novel contribution to bioacoustic representation learning, and also encourage the authors to consider updating the manuscript to reflect the points raised in the rebuttal.